# Disentangling the oceanic drivers behind the post-2000 retreat of Sermeq Kujalleq, Greenland (Jakobshavn Isbrae)

Ziad Rashed[1], Alexander A. Robel[1], and Hélène Seroussi[2]

[1]School of Earth and Atmospheric Sciences, Georgia Institute of Technology, Atlanta, GA, USA
[2]Thayer School of Engineering, Dartmouth College, Hanover, NH, USA

**Correspondence:** Ziad Rashed (zrashed3@gatech.edu)

**Abstract.** Ocean temperatures have warmed in the fjords surrounding the Greenland Ice Sheet, causing increased melt along their ice fronts, rapid glacier retreat, and contributing to rising global sea levels. However, there are many physical mechanisms that can mediate the glacier response to ocean warming and variability. Warm ocean waters can directly cause melt at horizontal and vertical ice interfaces, or promote iceberg calving by weakening proglacial melange or undercutting the glacier front.

Sermeq Kujalleq (also known as Jakobshavn Isbræ) is the largest and fastest glacier in Greenland and has undergone substantial retreat starting in the late 1990s. In this study, we use an ensemble modeling approach to disentangle the dominant mechanisms that drive the retreat of Sermeq Kujalleq. Within this ensemble, we vary the sensitivity of three different glaciological parameters to ocean temperature: frontal melt, subshelf melt, and a calving stress threshold. Comparing results to the observed retreat behavior from 1985-2018, we select a best-fitting simulation which reproduces the observed retreat well. In this simulation, the arrival of warm water at the front of Sermeq Kujalleq in the late 1990s led to enhanced rates of subshelf melt, triggering the disintegration of the floating ice tongue over a decade. The recession of the calving front into a substantially deeper bed trough around 2010 accelerates the calving-driven retreat, which continues nearly unabated despite local ocean cooling in 2016. An extended ensemble of simulations with varying calving threshold shows evidence of hysteresis in the calving rate, which can only be inhibited by a substantial increase in the calving stress threshold beyond the values suggested for the historical period. Our findings indicate that accurate simulation of rapid calving-driven glacier retreats requires more sophisticated models of iceberg mélange and calving evolution coupled to ice flow models.

## 1 Introduction

Observations indicate that many glaciers in Greenland have undergone rapid retreat over the last few decades. Sermeq Kujalleq in Kangia (hereafter SK, also known as Jakobshavn Isbræ) is a fast-flowing marine-terminating glacier in West Greenland, which has been the fastest flowing and largest contributor to Greenland ice discharge for the past several decades (Mouginot et al., 2019). SK has experienced considerable thinning and retreat with terminus velocities almost doubling from ∼6,700 m/yr in 1985 to ∼12,600 m/yr in 2003 (Holland et al., 2008; Joughin et al., 2014). As a result of this mass loss, SK was singularly responsible for 4% of the total rise of sea level in the 21st century. Prior to this rapid acceleration, SK was considerably slower, experiencing moderate thickening between 1991 and 1997 (Schweinsberg et al., 2017; Joughin et al., 2004). Before 2000,

SK had a floating ice tongue that provided some stability by buttressing the terminus (Echelmeyer and Harrison, 1990) and buffering frontal melt rates by acting as a heat sink for warmer fjord waters. The late 1990s saw warmer subsurface waters arrive in Disko Bay and the Illulisat Icefjord, leading to the collapse of the floating ice tongue (Holland et al., 2008), which is widely believed to have initiated SK's retreat and acceleration over the next 20 years. Prior to 1997, warmer water from the Irminger circulation was typically deeper than the sill, preventing significant exchange into the fjord; however, after 1997, the
rise of the Irminger water layer's upper interface to depths as shallow as 200 m enabled this warmer water to cross the sill and enter the fjord (Gladish et al., 2015).

Although warm subsurface waters are generally agreed to have triggered the most recent retreat phase of SK (Holland et al., 2008; Myers and Ribergaard, 2013), there is still debate about which physical processes were responsible for mediating the glacier response to ocean warming. Here we are primarily concerned in answering whether the following question: was
SK's recent acceleration and retreat caused by interannual variability in calving activity or by amplified melting of the ice front? Though there is some uncertainty over the extent to which warm waters penetrated the fjord beyond a submarine sill (Gladish et al., 2015), here we test the extent to which observed retreat could have been triggered by warm ocean waters and amplified by glaciological processes. The greatest increases in SK's terminus flow speed occurred in the summers of 2012 to 2015 (Khazendar et al., 2019) but initial warming (1-2°C) of Disko Bay fjord waters occurred a decade earlier, indicating
a delay in the SK flow response to warming ocean conditions. In contrast, SK flow speeds decreased concurrently with the cooling of Disko Bay water by 1.5 °C in 2017. Based on this relationship between glacier speed and fjord temperatures, it has been argued that enhanced melting of the terminus caused greater calving, retreat, and speed-up, particularly in summer when buoyant subglacial meltwater plumes should enhance circulation at the terminus (Khazendar et al., 2019). However, observational records indicate that previous intervals of enhanced fjord heat content prior to the 1980s did not result in the
same dramatic retreat (Slater et al., 2018), leading to the natural question: why did the most recent period of warming beginning in 1985 cause such a dramatic and unprecedented retreat? While water temperatures in Disko Bay are associated with melt and retreat at SK, it is still not clear whether this association indicates a causal relationship between enhanced terminus melt and observed thinning and retreat (Joughin et al., 2020), and whether the recent retreat is the direct result of oceanographic or glaciological factors. Alternatively, the strength of iceberg mélange and undercutting via frontal melt have been observed
to have strong control on calving frequency and style at SK and other glaciers (Joughin et al., 2004; Amundson et al., 2010; Cassotto et al., 2015; Luckman et al., 2015; Xie et al., 2019; Kajanto et al., 2023). Although this is not a universal feature among marine terminating glaciers in West Greenland (Amaral et al., 2020), the cliff-like geometry of SK's front may make it more susceptible to retreat in the presence of a weakening mélange. Thus, disentangling the drivers behind SK's response to warming ocean conditions requires distinguishing between retreat driven through ocean-induced melting due to increased
local temperatures and calving due to a weakened pro-glacial mélange.

Understanding the response of SK to warming ocean waters poses a difficult challenge due to the complex range of processes occurring at its interface with the ocean. Many processes that play a critical role in glacier stability (i.e., subshelf melting, calving, melt undercutting, mélange buttressing) remain poorly understood, despite recent advancements in high-fidelity glacier models (Benn et al., 2017; Slater et al., 2021; Wheel et al., 2024). Furthermore, the process of acquiring the necessary obser-

vations to constrain model parameters becomes challenging due to the presence of icebergs in winter seasons. Observational records of environmental variables such as mélange density (Kim et al., 2024; Wehrlé et al., 2023), detailed calving event catalogs, and calving front geometry are difficult to collect because icebergs act as physical barriers to oceanographic vessels, especially near the glacier front, where they are most dense. To avoid this, simplified parameterizations that relate calving rates to glacier stress and geometric conditions are used in many ice sheet models (Benn et al., 2007), but do not always capture the complex interactions between glacier and ocean state. Additionally, parameterizations that may perform well in describing one glacier might not perform as well at other glaciers (Amaral et al., 2020). Here, we use the rapid retreat and complex evolution of SK in recent decades as a natural experiment to better understand the uncertainties and shortcomings in simple parameterizations of ice-ocean interactions.

In this study, we simulate the historical evolution of SK from 1985 to 2018 using the Ice-Sheet and Sea-Level System Model (Larour et al., 2012). We perform a large ensemble of simulations of SK retreat through perturbation of three sensitivity parameters that control three processes which directly influence its retreat: subshelf melt, melt at the calving front, and calving threshold modulated by mélange rigidity. We compare model simulations to the relevant period of the observational record by scoring their ability to reproduce observed calving front geometry. We investigate the trade-off between different processes in driving SK's temporal and spatial sensitivity to melt- and stress-based mechanisms of mass loss and highlight the possible mechanisms most likely responsible for observed retreat. We emphasize that our goal here is not to accurately simulate mélange, but to offer a scenario in which disentangling the causes of SK's retreat can be accomplished with the consideration of long-term changes in mélange buttressing of the terminus through modification of the calving threshold parameter.

In section 2, we lay out the methodology for simulating SK's evolution from 1985-2018. We describe how simulations are initialized in the Ice-Sheet and Sea-Level System Model and the specific data used to recreate the state of SK in 1985. We then explain how melt and calving processes are parameterized with respect to ocean forcing and how we design our ensemble to determine which parameter combination results in model states that most closely matches observational data. In section 3, we present our model ensemble results and highlight key relationships between model parameters that best fit observations. In addition, we analyze the timing and extent of retreat within the model with the best match to observations. In section 4, we discuss implications for SK's future evolution given its current state. We also contextualize our findings in the context of Bondzio et al. (2018)'s study and use it as a control to compare against in assessing relative contributions of melt and calving to SK's evolution.

## 2  Methods

### 2.1  Model Configuration

We use the Ice-Sheet and Sea-Level System Model (ISSM) to simulate the retreat of SK from 1985 to 2018. ISSM is a state-of-the-art thermomechanical ice sheet model that has been used to simulate the evolution of glaciers and ice sheets from catchment to continental scales (Larour et al., 2012). Our modeling approach draws on some aspects of the configuration of a previous SK modeling study by Bondzio et al. (2018), with some key enhancements detailed later in this section. The domain

of our simulations includes the fast-flowing parts of the SK catchment and extends upstream deep into the SK catchment area. Using ISSM's built-in bidimensional anisotropy mesh generator, the domain mesh is refined using a metric based on the product of bedrock slope and surface velocity. The resulting mesh elements range in length from 400 m at high-velocity locations closer to the calving front to 4 km at lower-velocity locations deeper within the catchment area. A 2D shelfy-stream approximation (SSA; Morland and Zainuddin, 1987; MacAyeal, 1989) is used to simplify the three-dimensional flow equations as vertical gradients in velocity are relatively small and basal sliding is the dominant contributor to ice velocity at SK. The SSA approximation greatly reduces the computational expense of simulating marine-terminating glacier evolution and thus enables the large ensemble of simulations in this study. A linear-viscous Budd sliding relation (Budd et al., 1984) is used to relate basal shear stresses to basal speed. To obtain effective pressures, we first use a 1985 DEM (Korsgaard et al., 2016) and a 2009 DEM (Morlighem et al., 2017). To fill in the gaps in the 1985 surface, we use the 2009 DEM and apply a height offset proportional to the rate of surface height change in 2009. We follow a similar process in initializing the velocity field but instead use a velocity offset proportional to the velocity ratio between 2009 (Joughin, 2015) and 1991 (Mouginot et al., 2019) to fill in the 1985 velocity gaps. We then deduce initial Budd sliding coefficients from the updated 1985 surface heights and velocities. Grounding line migration is modeled using a sub-element migration scheme which allows the simulated grounding line to evolve continuously through mesh elements and reduces the dependence on mesh resolution (Seroussi et al., 2014). We use a model time step of approximately 5.5 hours in order to capture rapid changes in ice front geometry while maintaining numerical stability and accuracy.

Along the domain boundary, ice velocities are set to observed 1985 ice velocities, and the corresponding ice thicknesses are kept constant. Although SMB in the region of interest changed during the time period considered, the variations in SMB are small relative to the variations in ocean temperature (Hanna et al., 2011). Furthermore, it has been shown that the SK region is weakly sensitive to SMB forcing within our time period of interest (Seroussi et al., 2013) and that oceanic energy fluxes to the glacier increased while atmospheric energy fluxes to the glacier remained relatively constant during the time period of interest (Wang et al., 2020). Following this, we use a spatially variable surface mass balance held constant in time, based on a multidecadal mean from the RACMO regional climate model (Ettema et al., 2009).

Migration of the calving front is simulated using a level set formulation (Bondzio et al., 2016), where the migration rate of the calving front ($\boldsymbol{w}$) is determined using the difference between ice velocity and frontal ablation rate,

$$\boldsymbol{w} = \boldsymbol{v} - (c + m_{\mathrm{fr}})\,\boldsymbol{n} \tag{1}$$

where $\boldsymbol{v}$ is the ice velocity at the calving front, and $c + m_{\mathrm{fr}}$ is the ablation rate. Ablation at the calving front is driven by two parameterized processes: iceberg calving ($c$ is calving rate) and direct melt of the calving front by heat flux from the ocean ($m_{fr}$ is frontal melt rate). In the level set approach, ice flow advects the calving front downstream and ablation mechanisms move the calving front upstream.

## 2.2 Stress-based iceberg calving threshold

In ISSM, iceberg calving rate is calculated using a tensile-stress based criterion (inspired by criteria based on the von-Mises principal tensile stress)

$$c = |\boldsymbol{v}| \frac{\sigma}{\sigma_{\text{thr}}}, \tag{2}$$

where $\sigma$ is the principal tensile stress, $\sigma_{\text{thr}}$ is a prescribed stress threshold, and $\boldsymbol{v}$ is the ice velocity at the ice front (Morlighem et al., 2016).

Calving-induced retreat of the glacier front is initiated once local principal tensile stresses exceed the stress threshold parameter ($\sigma_{\text{thr}}$). The stress threshold parameter can be thought to conceptually represent many material characteristics such as fracture toughness, grain-scale deformation, and ice strength, which have the ability to modify the propensity for calving events. We parameterize a linear decrease in calving threshold, $\sigma_{\text{thr}}$, with increasing fjord ocean temperatures ($\boldsymbol{T}$)

$$\sigma_{\text{thr}}(T) = \sigma_{\max} - \frac{T - \min(\boldsymbol{T})}{\max(\boldsymbol{T}) - \min(\boldsymbol{T})}(\sigma_{\max} - \sigma_{\min}). \tag{3}$$

In Bondzio et al. (2018), the calving stress threshold alternates between a low stress threshold in the summer seasons and a high stress threshold in winter seasons, effectively shutting off the calving in the winter seasons and allowing the calving in the summer seasons. However, they do not include interannual variations in the maximum stress threshold. In our experiments, The minimum stress threshold, $\sigma_{\min}$, corresponds to the stress threshold in the presence of the warmest temperature in the fjord temperature time series, $T$, and conversely the maximum stress threshold, $\sigma_{\max}$, corresponds to the stress threshold in the presence of the coldest temperature in the fjord temperature time series. The linear variation in calving stress threshold is a simplified realization of the effects of mélange weakening/strengthening on buttressing the calving front, which has been posited as a possible explanation for observations indicating mélange weakening and breakup in concert with ocean temperature seasonality, rather than atmospheric temperatures (Kehrl et al., 2017; Bevan et al., 2019; Joughin et al., 2020). We use a linear sensitivity to relate ice mélange strength to ocean temperature for a simple comparison with other linear sensitivities assumed in this study. Due to the fractured nature of near-terminus ice at SK, we set the minimum calving stress threshold, $\sigma_{\min}$, to 100 kPa, below the measured mechanical strength properties of laboratory samples of pristine ice, typically in the range of 0.1-1 MPa (Lee and Schulson, 1988; Petrovic, 2003). The maximum stress threshold, $\sigma_{\max}$, parameterizes the potential roles of rigid iceberg mélange and melt undercutting in modulating the relationship between fjord ocean temperatures and calving rate and we use it to set the sensitivity of calving activity to local ocean temperature. Within our ensemble, we use a $\sigma_{\max}$ range of 220-350 kPa to capture a large swathe of potential calving behavior. The temperature dependence of the stress threshold ensures that calving activity increases when the ocean in contact with the glacier is warmer and vice versa. Although other material properties and glacier processes may play a role in setting the propensity for calving (e.g., ice fracture toughness, surface melt), they cannot explain the timing of seasonal and multiannual changes in the calving style in SK (Joughin et al., 2008). In Section 4, we further discuss the shortcomings of such a simplified representation of the effect of mélange on calving.

## 2.3 Frontal and submarine melt

Ocean melt of the glacier calving front (referred to hereafter as frontal melt) also contributes to the glacier response to fjord temperatures. We compute the frontal melt rates using the empirical parameterization from Rignot et al. (2016):

$$m_{\mathrm{fr}} = \left( A\, h\, Q_{\mathrm{sg}}{}^{\alpha} + B \right) T^{\beta}, \tag{4}$$

where $Q_{sg}$ is the subglacial discharge that is taken as the decadal average of total runoff of the SK drainage basin according to RACMO 2.3 (Noël et al., 2015), $T$ is the thermal forcing of the ocean obtained using the depth-averaged fjord water temperatures from the water temperature in the Egedesminde Dyb in the coupled ocean and sea ice simulation provided by the Estimating the Circulation and Climate of the Ocean, Phase II (Menemenlis et al., 2008), and $h$ is the depth of the water column. A is a tuning parameter and B is a parameter that ensures that the heat flux does not vanish in the absence of melt water (Rignot et al., 2016). This empirical equation has been shown to be a good approximation of how frontal melt relates to ocean temperature and subglacial discharge at several glaciers in Greenland, based on in-situ observations and numerical simulations (Rignot et al., 2016). Since $Q_{sg}$ and $\alpha$ are sufficiently small and $\beta$ is sufficiently close to one, we assume that frontal melt is linearly proportional to thermal forcing to simplify the following analysis. We do not account for the effects of convective plume forcing on frontal melting because the sensitivity of frontal melting to subglacial discharge is lower than to ocean temperature (Xu et al., 2013), especially after the removal of the ice tongue.

Submarine melt on the floating portion of the glacier is the final process that we consider to contribute to the glacier response to fjord temperatures. We simulate the submarine melting process by assuming that the water column is stratified such that water at maximum depth is also the warmest water. We use this assumption to justify the warmest waters overcoming the shallow sill and infiltrating SK's fjord. Following this, we take the depth-average of the water column's temperature between the surface and maximum of bedrock and sill height (-250 m). We fill the data gap between 1986-1992 by repeating the temperature time series between 1992-1996. The submarine melting rate is then computed using the simplified parameterization from Holland and Jenkins (1999):

$$m_{\mathrm{sm}} = -\rho_M c_{pM} \gamma_T (T - T_{pmp}). \tag{5}$$

where $\rho_M$ is the density of the mixed layer, $c_{pM}$ is the specific heat capacity of the mixed layer, $\gamma_T$ is the thermal exchange velocity and $(T - T_{pmp})$ is the difference between the temperature of the mixed layer and the melting point temperature at the base of the ice.

We then choose a parameter space of multipliers for melt mechanisms such that we encompass a large range of possible melting scenarios. For our frontal and submarine melt multipliers, $\alpha_{\mathrm{mf}}$ and $\alpha_{\mathrm{ms}}$ respectively, we chose a range of 0-4x the empirical parameterizations mentioned in equations (5) and (4). Following a series of experiments within different ranges, this range was selected to capture as much variability in simulation output while producing simulated retreats at least somewhat resembling the observed evolution of SK.

| Parameter | Value | Units |
|-----------|-------|-------|
| $A$ | $3 \times 10^{-4}$ | Unitless |
| $B$ | 0.15 | Unitless |
| $\alpha$ | 0.39 | Unitless |
| $\beta$ | 1.18 | Unitless |
| $\rho_M$ | 1023 | $\mathrm{Kg\,m^{-3}}$ |
| $c_{pM}$ | 3974 | $\mathrm{J\,K^{-1}\,Kg^{-1}}$ |
| $\gamma_T$ | $1 \times 10^{-4}$ | $\mathrm{m^2\,s^{-1}}$ |
| $T_{pmp}$ | -1.85 | $^\circ$ C |

**Table 1.** Overview of all relevant melt parameters.

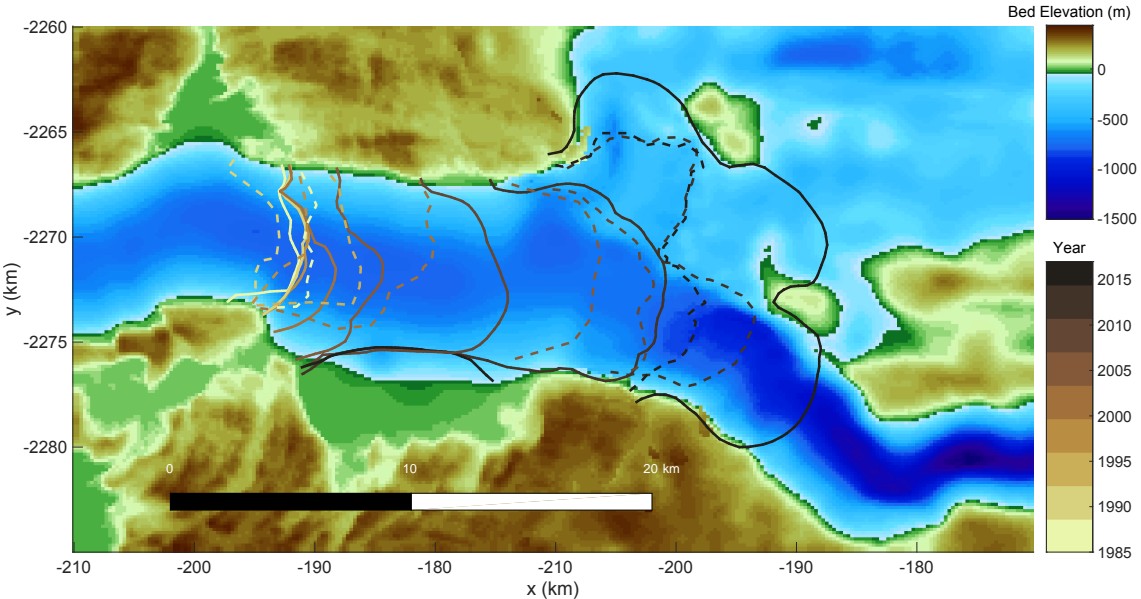

**Figure 1.** Observed and simulated calving fronts from our highest scoring simulation at SK from 1985-2018. The solid lines denote outputs from our best fitting ensemble member and the dashed lines indicate the observed positions of SK's calving front with the color corresponding to the year of observation.

## 2.4 Model-Observation Mismatch Score

We choose to plot simulations with a max stress threshold varying over 220-350kPa with an interval of 10 kPa to ensure that we capture both potentially vigorous and negligible calving activity. Each simulation is scored on the basis of its ability to match the observed position and geometry of the glacier calving front. We use a historical catalog of SK's calving front geometry (i.e., a 2-D curve) obtained from observational records which includes multiple snapshots of calving front geometry derived from

Landsat 5–8, ERS-1 and 2, and TerraSAR-X (Moon et al., 2014) satellite imagery (Figure 1). For each point in time in which we have an observation, we pair each simulation's observed terminus geometry with the nearest-in-time (always within 5 days) modeled terminus geometry from ISSM. For each observation-model pair, we calculate the area between the modeled and observed terminus geometry. The resulting mismatch vector contains the difference in geometric area between modeled and observed geometries, and we take the root mean square of this vector to assign a score. This is different from the approach of Bondzio et al. (2018) which only considers the center line position of the SK calving front; considering the entire calving front geometry allows for more accurate tracking of the glacier when the front bifurcates, as it did in 2006. Additionally, while we do not weight area differences to account for changing observational density in time, almost all of the retreat of SK occurs during a time period (post-2000) when observational density is high. Thus, the mismatch score is unlikely to be strongly dependent on observational density. Using this scoring method, we can accurately capture changes in calving front position and shape, and we are only limited by the resolution of the observational records and model meshing. Convergence studies indicate that at our chosen model time step, potential errors due to the mismatch in timing between model and observation contribute negligibly to the overall scored metric.

## 3  Results

The speed and timing of the simulated retreat of SK vary depending on the sensitivities of the calving threshold, submarine melt, and (to a lesser extent, discussed in the following section) frontal melt to the local ocean temperature, but there are some commonalities between simulations in our perturbed-parameter ensemble. Figure 1 shows the observed calving front positions (thick lines) and the results of one simulation with the best correspondence to observations (dashed lines). Generally, the simulated calving fronts of SK pause at locations where there are geometric pinning points such as bedrock peaks and fjord narrowings. However, the length of time that the simulated glacier remains at such pinning points varies, due to the interplay of calving and melting. Initially, the ice tongue of SK stabilized the glacier front by buttressing inland ice. As in observations (Motyka et al., 2011), simulations indicate that intensified submarine melting due to warming of fjord waters starting around 1997 thinned the ice tongue and weakened its buttressing capacity. The observed weakening and subsequent disintegration of SK's ice tongue resulted in dynamic thinning of the terminus and an acceleration in retreat. Similarly, most of the simulations in our perturbed parameter ensemble yield submarine melting rates which peak between 1995 and 2000, as warm water entered Illulisat Icefjord via Disko Bay. The simulated disintegration of the ice tongue and subsequent front retreat leads to the bifurcation of the calving front into two branches and exposure of a much larger frontal area to warm ocean waters (Figure 1). The response of SK's two branches is not homogeneous owing to large differences in their bed topography and fjord geometry. Ice fluxes are greatest along the southern calving front where the bedrock is deeper and upstream topography is characterized by more extensive retrograde slopes (Figure 1). The combination of a thicker and steeper glacier terminus and a deepening grounding line bed slopes facilitate ice loss via calving. Thicker ice results in a greater overburden pressure which, at the grounding line, is counterbalanced by hydrostatic pressure and buttressing stresses from floating ice and mélange. Thus, after removing the ice tongue, tensile stresses grow rapidly along the southern front (Figure S1) and calving quickly

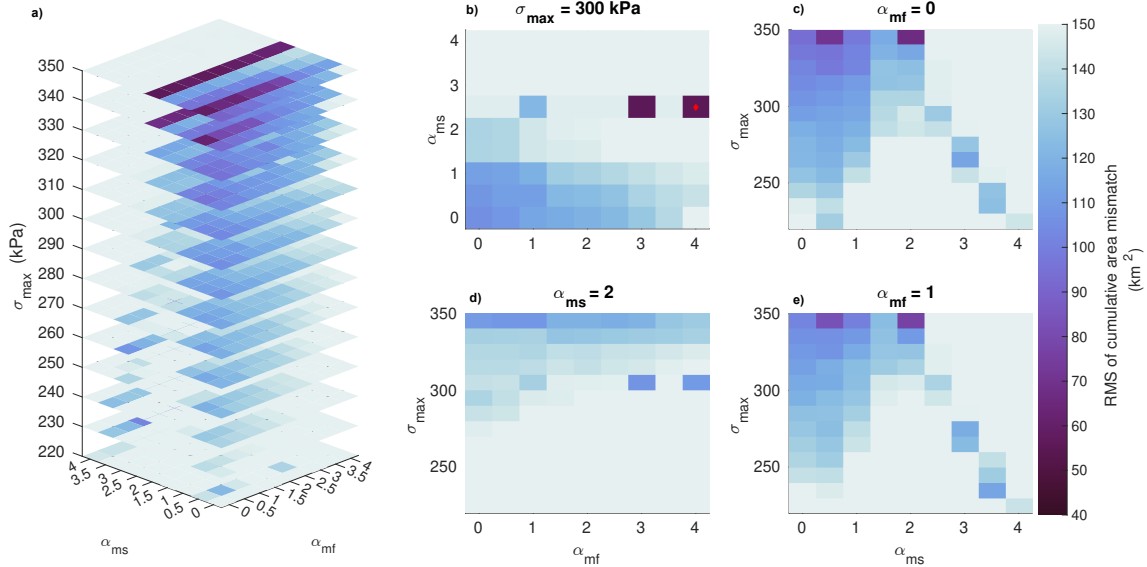

**Figure 2.** Parameter combination sweep of relevant melt values. Darker colors indicate a smaller RMS of cumulative area mismatch between modeled and observational records of SK's calving front position. Panel a displays the RMS of cumulative area mismatch for specified maximum stress levels. Panels b-d show respective RMS slices for $\sigma_{\mathrm{max}} = 300\mathrm{kPa}$, $\alpha_{\mathrm{mf}} = 0$, $\alpha_{\mathrm{ms}} = 2$, and $\alpha_{\mathrm{mf}} = 1$. The red diamond in panel b indicates where our best-fit parameter combination lies in parameter space.

becomes the dominant ice loss mechanism (Figure 3). The northern branch of SK experiences much less retreat, similarly to

what observations show (Figure 1). The shallow bed topography and an abundance of pinning points constrain the upper branch from rapidly retreating following ice tongue disintegration.

### 3.1  Perturbed-parameter ensemble of SK retreat simulations

In our large perturbed-parameter ensemble, multiple simulations were able to achieve nearly equivalent matches to the observed retreat of SK (Figure 2). There appears to be a minimum achievable match to observations (with RMSE of approximately 50

230  km$^2$ over the simulation period) related to observed changes in calving style unrelated to ocean temperatures, which we discuss in more detail in section 4. We also expanded our ensemble extent within parameter space beyond the ranges plotted in Figure 2 to verify that the best fitting simulation was indeed the best fit. We do not plot simulations from this greater ensemble due to the lack of glacier retreat under a sufficiently large stress threshold regime. The best scoring simulations all occur within a region of parameter space where submarine melt rates are close to what would be predicted by Equation 5 without the need for an

ad-hoc multiplier and maximum stress thresholds relatively high, as plotted in Figure 2. The cold-ocean maximum of calving stress threshold in this region of parameter space is 250-400 kPa, generally much lower than suggested by laboratory studies, and near the low end of the range of observationally derived values for fractured glacier fronts (Vaughan, 1993; Choi et al.,

2018). Outside this region of parameter space, model-observation mismatch scores are consistently much worse, indicating an implicit role for rigid mélange in butressing the calving front and preventing calving.

The simulations best matching observations generally require submarine melt to be slightly more sensitively dependent on ocean temperatures ($\alpha_{\mathrm{ms}} > 1$) than suggested by the parameterization (Rignot et al., 2016). However, there is a notable trade-off between the stress threshold and submarine melt, such that simulations with higher cold-ocean stress thresholds (i.e., less calving in coldwaters) also need higher submarine melt rates to achieve reasonably low RMS (Figure 2b). Early in the simulation (1985-2000), the diminished calving of the ice tongue must be compensated for by amplified submarine

melt to accurately simulate the timing of the ice tongue collapse. Consequently, the greatest mismatch between models occurs following the collapse of the ice tongue, coinciding with the onset of calving-dominated retreat (Figure 3b).

    Though the absolute best-fitting simulation requires high sensitivities of front melting to ocean temperatures ($\alpha_{\mathrm{mf}} = 4$), there are several simulations with very similar RMSE requiring little to no front melt at all to fit observations (Figure 4). In these simulations, the greatest mismatch occurs following ice tongue collapse which is when calving becomes the dominant mode

of ice loss. It is also by this point that thermal forcing from warmer waters contributes less to SK's dynamic response. If the stress threshold is sufficiently low, retreat following ice tongue collapse is controlled by bed topography and calving in the southern trunk. Conversely, retreat is controlled by submarine melt when stress thresholds are considerably greater than tensile stresses generated at the glacier front. Changing frontal melt rates predicted by the Rignot et al. (2016) parameterization by modifying $\alpha_{\mathrm{mf}}$ does not significantly change the behavior of the model or improve our combination of best fit parameters. The

weak dependence of RMSE in Figure 2 to the sensitivity of frontal melt to ocean temperature indicate that, at least for SK, calving and submarine melt control the speed and timing of glacier retreat.

## 3.2   Best-match simulation

As observed from 1985 to 2000, SK maintained a floating tongue that at some locations extended more than 10 km from its grounding line. As long as a floating ice tongue existed, observed surface ice flow velocities exhibited very little seasonal

variability (Echelmeyer and Harrison, 1990), indicating a glaciological state in which calving and melting are consistently balanced by surface accumulation and ice flow upstream. SK's ice tongue acted as a buffer for retreat by transmitting buttressing back stress from the Illulisat Icefjord walls to the grounding line. Following the influx of warmer ocean water into Disko Bay in 1997, local fjord water temperatures abruptly warmed which led to the disintegration of SK's ice tongue over several years and the subsequent thinning and accelerating retreat of the newly exposed terminus (Joughin et al., 2020).

Similarly, in our best-fitting simulation, the increase in ocean temperature beginning in 1997 (Figure 3c) causes an increase in submarine melt fluxes, which is then followed by the simulated ice tongue thinning and retreat. Submarine melt initially dominates ice loss, peaking as ocean temperatures reach their maximum in 2000. Following this peak, submarine melt fluxes slowly decrease, albeit at a rate faster than ocean temperatures decrease because of the decreasing area of the floating ice tongue. During this period, the base of the ice tongue steepens, which causes a subsequent increase in driving stress and

extensional stresses, which promotes calving activity. Although we expected to see a similar increase in calving fluxes due to the decreased stress threshold under warm water conditions, we did not see an increase until 2010 (Figure 3a). This is most

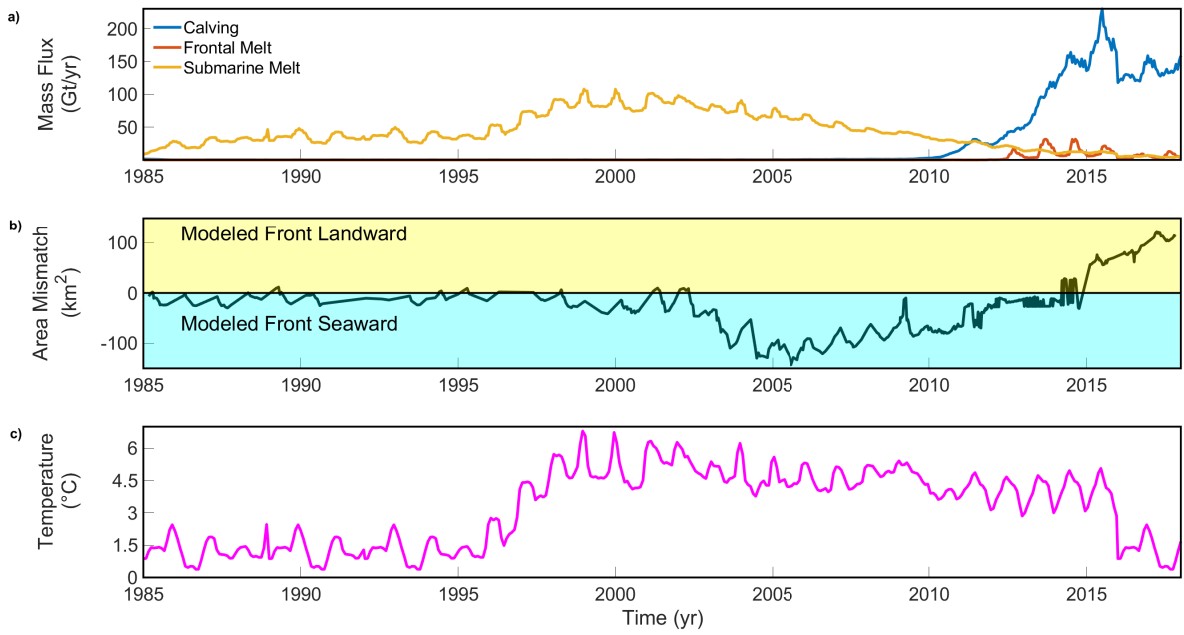

**Figure 3.** Best fitting model flux contribution and performance. (a) Mass flux contribution of relevant ablation mechanisms to the evolution of SK's calving front. (b) Observed area change time series vs modelled area time series. (c) Estimated Disko Bay fjord temperatures sourced from ECCO2 (Menemenlis et al., 2008).

likely due to the response timescale associated with the geometric adjustment of the floating ice tongue and corresponding stress state of the terminus, which depends more on the instantaneous glaciological state rather than the oceanic state. It should also be noted that during this period of retreat, SK's front is still far enough downstream that it has not begun to retreat into its

two separate branches (Figure 1). Throughout this period, SK's grounding line sits on a slightly prograde bed slope and also has not yet begun rapid retreat. Thus, this period is marked by a relatively slow retreat of the floating calving front, driven by submarine melt, which sets the glacier up for further retreat as persistently warm water continues to reach the glacier front.

During the period from 2000-2012, in our best-fitting simulation, SK's retreat begins to accelerate. In this phase of the retreat, the change in SK's geometry begins to play an important role in setting the calving rate at the terminus as the front

retreats onto a much narrower and deeper bed (Figure 1). From 2000-2005, there is a decrease in submarine melt flux caused mostly by the reduction in floating ice tongue area, and to a lesser extent by the relative decrease in ocean temperature forcing. By 2005, SK's calving front had retreated enough to lie across the upper and lower branches, which introduced a greater variability in bed topography along the front (Figure 1). The greatest sustained discrepancy between modeled and observed glacier geometries occurs over the 2005-2010 period, as our modeled terminus lags downstream of the observed terminus

(Figure 3). The mismatch in geometry is largely amplified by the differences in calving rate as there is a trade-off between

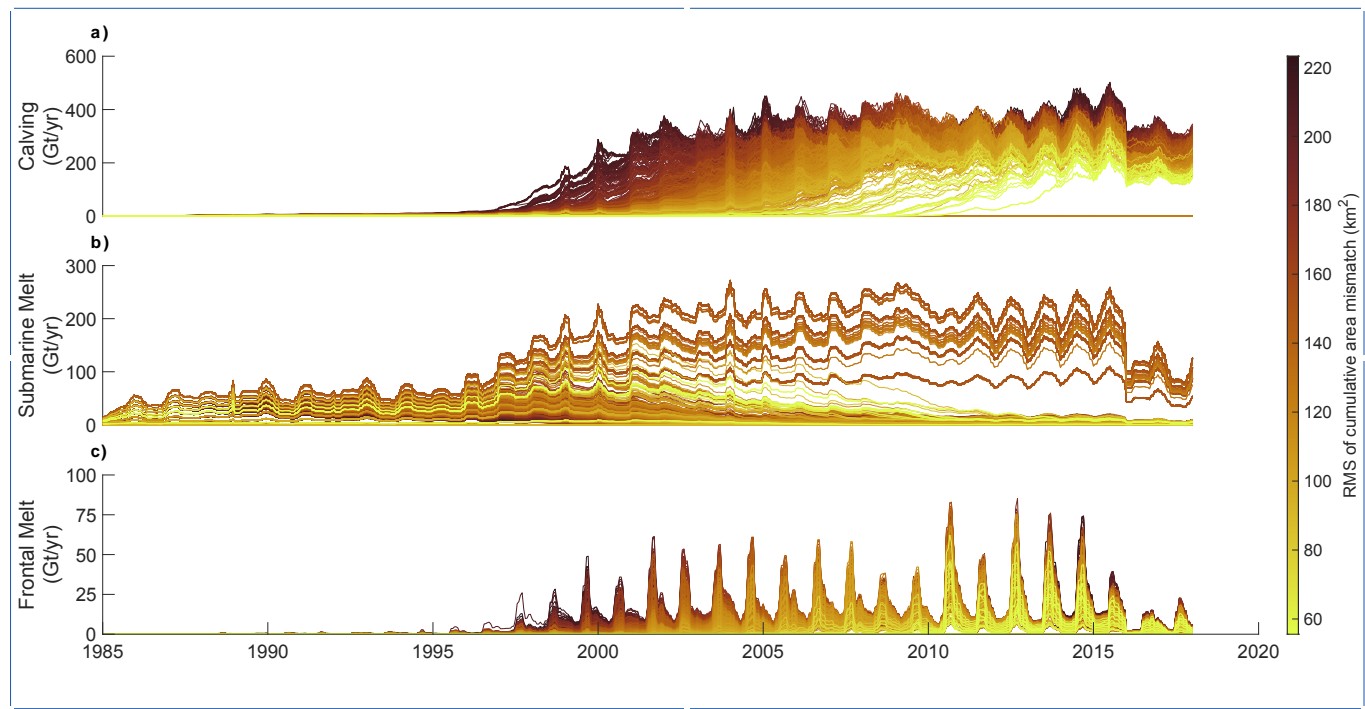

**Figure 4.** Mass ablation rates of all initial ensemble members color coded by their RMS of cumulative area mismatch. Panels a-c show how the magnitude of the stress threshold and melt multipliers relate to the overall RMS. Note how the best scoring models tend to favor lower calving stress activity.

capturing accurate calving rates after 2010 and insuring minimal mismatch between 2005 and 2010 as the dominant mode of ice loss changes. Throughout this period, we observe the transition to a calving-dominated retreat owing to geometric changes at the calving front. The combination of a retrograde bed and a steep ice cliff front geometry promotes intensified calving activity along the lower branch of SK. By 2010, SK's ice tongue has been completely removed and the calving front is a steep ice cliff. The time at which model ensemble members reach this transition point would be the greatest source of variability in overall error between modeled and observed calving fronts.

After 2010, the best-fit model simulation includes a rapid acceleration in calving rate and a better fit between modeled and observed calving fronts. Between 2010 and 2015, the dominant retreat mechanism transitions from submarine melt dominated to calving dominated with melt accounting for a greater ice flux at the beginning of the period and calving at the end of the period (Figure 3a). This shift is attributed to the retreat of the grounding line into a deeper, retrograde-sloping bed.. The transition of SK to a calving-dominated retreat after 2010 (Figure 3), and the lack of substantial re-advance during the brief cooling between 2016-2018 suggests the possibility that our simulated SK has undergone a hysteretic change to irreversible calving-dominated retreat as discussed further in the next section. Due to the strong topographical control on retreat rate, we expect that once the lower branch's calving front retreats past the over-deepened trough, a rapid acceleration of retreat is very

likely regardless of climate forcing (Kajanto et al., 2020). Further retreat is largely driven by calving fluxes as floating ice area considerably decreases and submarine melting becomes less influential on glaciological state. We also note that frontal melt fluxes begin to overtake submarine melt fluxes during this period due to increased exposure area of the ice front to warm waters, but this increase is still an order of magnitude smaller than the relative increases in calving fluxes.

### 3.3 Potential hysteresis effects of ice mélange

In our best-fitting simulations, calving rate at SK rapidly accelerates in concert with retreat into a deep trough on the southern flank of the glacier catchment. The glacier bed in this trough further deepens 20-30 km upstream of the 2018 calving front position, raising the specter of hysteresis to permanently high calving rates, regardless of future climate forcing. However, other studies have raised the possibility that increased calving will produce a thicker ice mélange (Xie et al., 2019; Cassotto et al., 2015) which could inhibit calving. As described in section 2, our simulations do not account for this potential negative feedback, and assume that calving rate is only sensitive to ocean temperatures and glacier front geometry.

To simulate the potential for hysteresis effects associated with calving, we continue the best-matching simulation from the large ensemble described in the previous section in a series of simulations with an increased calving stress threshold until 2100. To do this, we run an ensemble of simulations continuing the best-fit model from 2018-2100 where maximum calving stress threshold is set to a multiple (1-2.5x) of the calving stress threshold at 2018 and held constant until 2100. Similarly, we keep the temperature forcing constant at 2018 values until 2100 such that frontal and submarine melting rates remain constant. Figure 5 shows that the resulting simulations exhibit two types of behavior. When the calving stress threshold is maintained at 260 kPa, the stress threshold value given by the best-fitting simulation in 2018, calving rate continues to increase, peaking around 2050 at rates more than twice 2018 rates. This behavior is shared with a grouping of ensemble members (260-437 kPa) or (1-1.67x 2018 value of $\sigma_{max}$) and consistently results in retreat far into the ice sheet interior by 2100. Ensemble members with a stress threshold above 437 kPa ($\sim$1.67x) stabilize, but do not readvance. In these simulations, the calving front stabilized in the deepest trough along the flowline, and the glacier experienced gradual thickening. We note here that such calving stress thresholds were not attained at any point during the historical time period used to calibrate the stress threshold i.e., they are above the cold ocean $\sigma_{max}$ found for all best-fitting simulations in the parameter sweep described in the previous section. Thus, even if ocean temperatures returned to the coldest values achieved during the historical period over the next 80 years, the rapid acceleration of calving and retreat would likely continue unabated under the assumption that calving threshold is only determined by ocean temperature and the stress state at the calving front. this runaway retreat highlights that simulations which represent calving in such a simplistic way will tend to produce irreversible retreat, which may not be the case if mélange is realistically represented. Importantly, almost all current ice sheet models that use stress-based calving parameterizations hold such parameters constant in time.

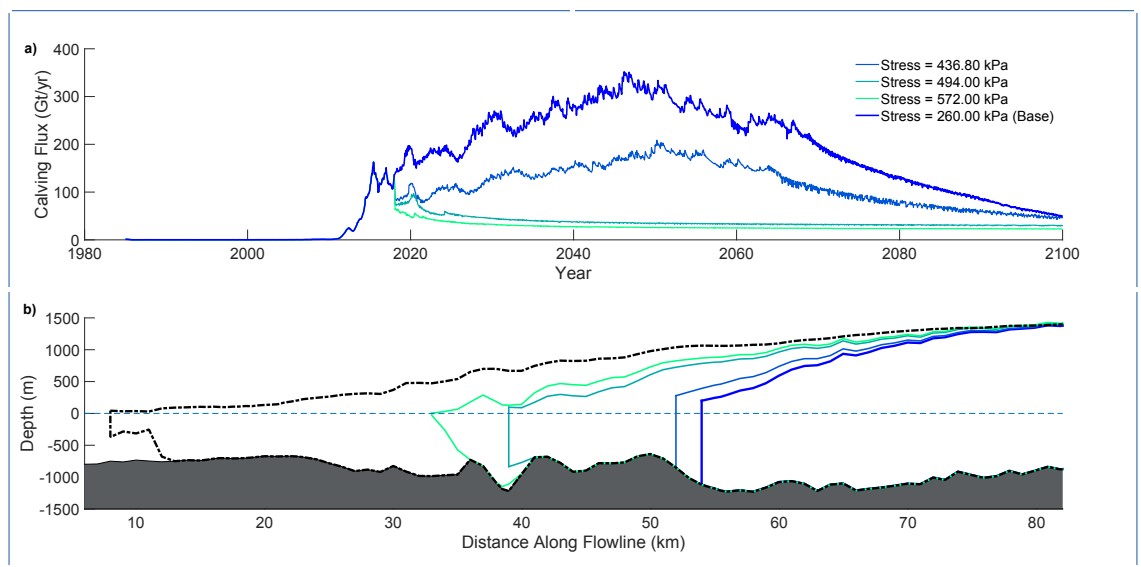

**Figure 5.** Top Panel (a): Calving fluxes of 1985-2100 run ensemble members. Bottom Panel (b): Along flowline profiles of ensemble members at year 2030. The initial 2018 profile is denoted with a thick black dashed outline and sea level is denoted with a blue dashed line.

## 4    Discussion

In this study, we have investigated the drivers behind SK's retreat from 1985-2018. By considering only processes related to ocean forcing, we are able to reasonably reproduce the observed evolution of SK and analyze the mechanisms responsible for its retreat before and after ice tongue collapse. Our best-fitting model simulations match those sparse observations that do exist but also allow us to understand the physical mechanisms that drove the observed retreat in a physically self-consistent manner. Thus, we can paint a fuller picture of the dynamics responsible for SK's evolution.

We noted two distinct behaviors in glacier evolution in the simulations beyond 2018 (Figure 5). The first is stabilization of the calving front retreat on a prograde slope following rapid retreat. The second is characterized by a rapidly accelerating retreat which extends 10's of km into SK's interior followed by a slow down of retreat. The timing of rapid retreat and arrest varies marginally between ensemble members. By 2021, ensemble members which experience a slow down in retreat stabilize near the bedrock peak approximately 40 km along the flowline (Figure 5b). This further reinforces the notion that SK's current retreat is controlled by bed topography, with the stress threshold (and all the factors which determine it) influencing the amount of time it takes to reach the rapidly calving state. This behavior is indicative of threshold behavior, wherein a small change in a parameter (in this case calving stress threshold) will lead to either a stable and slightly advancing SK or a rapidly retreating and unstable SK.

Additionally, throughout the parameter space considered in this study, there is a clustering in calving front positions of 2018 (Figure 6). On one end, there is a cluster that do not experience ice tongue loss and subsequently do not experience rapid calving. These simulations are characterized by either lower calving stress thresholds or melting multipliers (Figure

S2). Although calving front locations are more sensitive to calving thresholds, insufficiently strong ocean melt is unable to generate a rapid removal of the ice tongue, which is a necessary condition to produce mechanical imbalances at the calving
front and subsequent retreat. Within these simulations, the calving front is quickly prevented from retreating and remains at the same position. On the other end, we see simulations that quickly lose their ice tongue and subsequently accelerate faster than observational records. In this set of simulations, the calving stress threshold plays a greater role than melting in setting the rate of retreat as the grounding line retreats onto deeper bedrock and the glacier front experiences greater mechanical stresses. When letting these simulations continue beyond 2018, we note that rapid calving continues along the southern trunk of SK, but
stops across the northern trunk, owing to the presence or lack of steepening bed slopes, respectively. There are no simulations with calving fronts stabilizing along a position between the observed calving front in 2018 and a few kms inland from the initial calving front in 1985. The most likely reason for this lies in our simplified parameterization of a dynamic ice mélange. In the absence of mélange, the behavior of the SK calving front as modeled here can be characterized either by rapid and vigorous retreat or terminal stability. This leads to a bimodality in model response, while, in reality, ice mélange acts as a stress buffer
in response to rapid calving.

In designing simulations that can be used to disentangle the drivers of SK retreat, certain simplifications were necessary, which limit the applicability of this study to other glacier settings. In order to assess the main drivers behind SK's retreat, certain assumptions were made in our simulations. We do not use higher-order approximations for the glacier momentum balance (i.e., Full Stokes), instead relying on the SSA in order to reduce computational expense, enabling a large ensemble of
simulations while maintaining an accurate representation of grounding line dynamics. A direct consequence of simplifying the flow equations is the overestimation of the basal drag coefficient near the grounding line (Morlighem et al., 2010). On the other hand, Bondzio et al. (2017) showed that a linear-viscous Budd sliding relation (Budd et al., 1984) and coefficients captured SK's velocity well.

Calibrating multiple model parameters (in this case $\alpha_{mf}$, $\alpha_{ms}$ and $\sigma_{max}$) allows us to produce simulations which fit observed
retreat of SK reasonably well. In Bondzio et al. (2018)'s study, a novel large ensemble approach was used to forecast SK behavior under a collection of parameter combinations of $\alpha_{mf}$, $\alpha_{ms}$, and $\sigma_{max}$, which were kept constant over time with a rectified seasonal cycle in calving threshold. We expanded on this work by dynamically changing calving propensity through inter-annual change in ocean temperature. Our modified approach produces modeled calving fronts which closely match observed calving fronts without requiring frontal melt to be multiple times higher than suggested by empirical parameterizations
(Rignot et al., 2016). However, ultimately such an approach is limited in its ability to explain the role of physical processes not included in our modeling system and how they may evolve outside of the historical sample of satellite observations of SK. The extended simulations described in Section 3.3 indicate that the only way to arrest the future retreat of SK in our modeling system is to increase the calving threshold to values never attained during the historical period (Figure 5a). We recognize that it may be possible that SK is indeed engaged in a runaway retreat that will not be arrested by any mechanism in the next century.
However, to allow for the possibility that SK retreat could slow down or stop in the future, other processes that play a role in modulating calving must be simulated. SK's potential to generate large and dense ice mélange fields (Cassotto et al., 2015) and its strong dependence on calving front conditions (Bondzio et al., 2017) suggests that a strong enough ice mélange, potentially

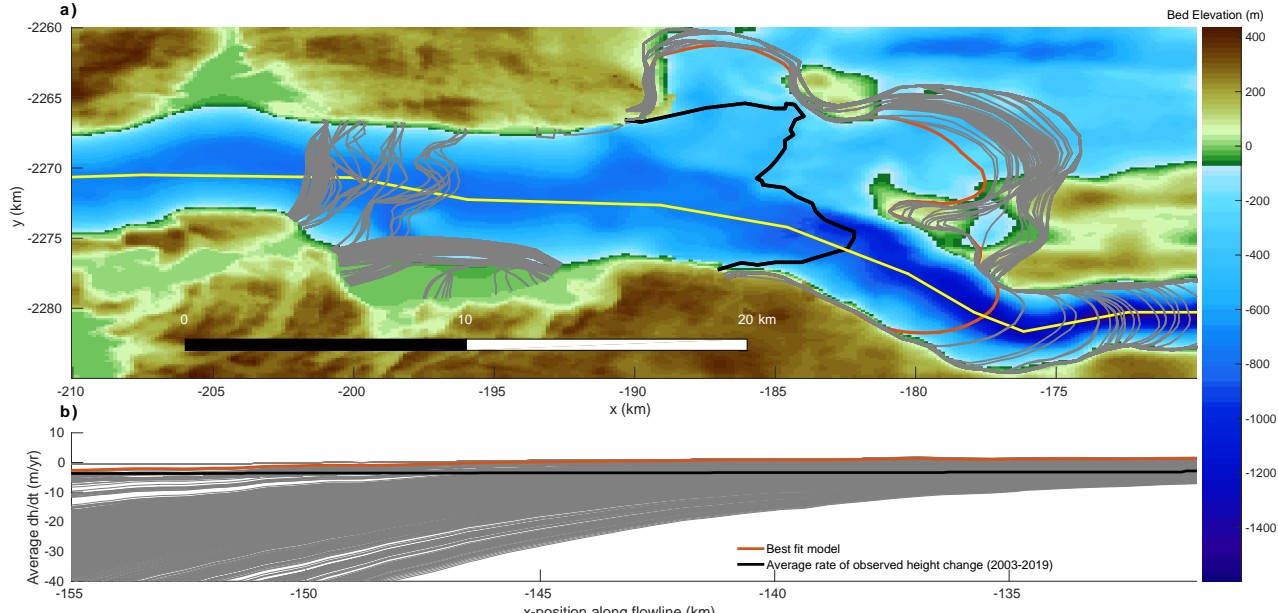

**Figure 6.** Top panel (a) depicts the average rate of surface height change for all ensemble members along a flowline (denoted in(b)) along the southern SK branch. The average rate of observed height change is obtained from ICESat and ICESat-2 data from 2003-2019 (Smith et al., 2020). Bottom panel (b) displays the 2018 calving front positions for all ensemble members. Additionally the best fit model calving front is highlighted with a dark orange and the observed 2019 winter calving front Smith et al. (2020) with black. The thin yellow line indicates the flowline on which we calculate surface height change.

generated through vigorous calving, may be able to suppress runaway calving behavior. Simulating such a feedback would require a dynamic model of ice mélange coupled to the ice sheet and the ocean below. No mélange model currently exists which fits these requirements, though prior efforts have produced useful parameterizations (Vaňková and Holland, 2017; Pollard et al., 2018; Amundson and Burton, 2018; Schlemm and Levermann, 2021b), uncoupled models (Amundson et al., 2025) and very computationally expensive tools unfit for coupling to ice sheet models (Robel, 2017; Burton et al., 2018). While our goal in this study was to provide a different perspective on how ocean-temperature-based processes could modulate calving in the context of SK, our findings indicate that a fully capable and coupled model of mélange is a pre-requisite for any attempts to accurately model SK's future evolution.

## 5 Conclusions

In this study, we conducted numerical simulations of Sermeq Kujalleq using ISSM to disentangle the relative importance of different mechanisms in driving the retreat of SK from 1985-2017. Using a large ensemble of parameter-perturbed simulations, we explored a wide parameter space of calving and ocean melt parameterizations and compared calving front positions to key observations to score a given simulation on how well it could match observations of retreat. We found that submarine

melting and calving both played critical roles in the timing and magnitude of SK's retreat. Specifically, we note that intensified submarine melting due to a potential intrusion of anomalous deep-water temperatures was necessary to instigate rapid retreat by melting SK's ice tongue. Following the loss of the ice tongue, calving became the dominant retreat mechanism, due to the exposure of a tall calving front with correspondingly high stresses exceeding the calving threshold. Calving rates rapidly
increased as SK retreated into deeper waters and therefore experienced greater tensile stresses along its calving front.

A central finding of this study is that the ability of ice mélange to buttress SK's calving front increases in importance as SK's calving front rapidly retreats onto deeper beds. We tested this by extending simulations until 2100 and analyzed the potential for increased calving stress threshold to arrest further retreat of SK. Our simulations reveal that a sufficiently robust ice mélange could suppress calving activity during SK's most vigorous calving phase. The loss of SK's ice tongue subjected the
405 calving front to greater tensile stress, which consequently amplified the importance of ice mélange importance in modulating calving rates. We hypothesize that the influx of warm waters into the Illulisat Icefjord facilitated downstream movement and fragmentation of ice mélange. Further testing of this hypothesis is required, but we have shown here that calving variability, in its current state, is the dominant control on SK's evolution. Additionally, simple parameterizations of mélange such as those employed here and other studies attempting to quanitfy mélange butterssing effects (Schlemm and Levermann, 2021a; Parsons
et al., 2024) are ultimately meant as upper bounds on glacier-mélange interactions since they do not represent the true rheology of mélange nor its two-way with the calving front. This finding emphasizes the importance of future development of dynamic models of ice mélange evolution that can be coupled to models of vigorously calving glaciers .

*Code availability.* Relevant run and analysis scripts required to produce results found at a persistent Zenodo repository located at https://doi.org/10.5281/zenodo.11176342.

*Author contributions.* All author help conceive the study and contributed to the manuscript. ZR conducted model simulations, post-processing and analysis.

*Competing interests.* The contact author has declared that none of the authors has any competing interests.

*Acknowledgements.* Thanks to Blake Castleman for early work on simulations, and Johannes Bondzio for answering questions on ISSM configuration. Vincent Verjans and John Christian provided useful comments during the completion of this work. This research and authors
were supported by funding from a JPL Strategic University Research Partnership (SURP) SP20007. Z. Rashed and A. Robel were also supported by NSF OPP Grant 2025692. H. Seroussi was also supported by NSF NNA Grant 2127246.

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
