# Peer review of "Disentangling the oceanic drivers behind the post-2000 retreat of Sermeq Kujalleq, Greenland (Jakobshavn Isbrae)"

_EGUsphere, 2024_

## Referee Comment (RC2)

[referee-annotated manuscript omitted]

---

## Author Comment (AC1)

**Overview**

This study seeks to determine which mechanism of frontal and submarine melt and ice-ftont calving primarily drove the retreat of Sermeq Kujalleq glacier between 1985 and 2018. Ensemble modelling was performed whereby the magnitude of previously published frontal and submarine melt rates were adjusted alongside the calving stress threshold of a tensile stress based calving law. The calving law was further adapted in an attempt to account for backstresses imparted by ice melange. Optimised parameters were then used to assess potential future behaviour of Sermeq Kujalleq up to 2100.

Although the question of whether oceanic melting or calving processes were the ultimate drivers of Sermeq Kujalleq's retreat is of great interest, there could be more detail given in methods used in the study in order to justify how physically meaningful the employed parameters are. Further, improvement to existing figures as well as the addition of further figures may help with the clarity of results interpretation and the conclusions drawn from the study. With further clarification added to address the comments below I believe this manuscript would be valuable to the community and recommend it for publication.

We would like to thank the reviewer for their insightful remarks. Below is our initial response, broken down by remark. Original review comments are shown in black, while our answers are shown in red.

Comments

Line 8: Are all the drivers coming from the ocean? There is no mention of atmospheric drivers. Can it be discussed in the methodology why atmospheric drivers are not considered here and the title of the paper should be more specific on this.

There is atmospheric forcing through SMB but it is held constant to the multi-decadal mean obtained from RACMO throughout the simulation. In Methods, line 90, we note that there exists much greater interannual variability in oceanic forcings than in atmospheric forcings. We will clarify this in text.

Line 49-50: 'the process of acquiring necessary observations…' Which observations? And why do icebergs make it more challenging?

Observations of environmental variables such as mélange density, detailed calving catalogues, geometry of calving front. Icebergs act as physical obstacles which may complicate data collection methods, especially near the glacier front. We will clarify why these observations are relevant and their associated difficulties more clearly in text.

Line 77: I'd find it informative to include a figure of the study area. Where are the fast flowing regions? Where is 'deep in the catchment area' and what are the flow speeds here?

We will include a supplemental figure showing simulated SK's velocity field from MEASURES data and make appropriate references when discussing flow speeds.

Line 78: How are different areas of the domain chosen for coarsening / refinement?

Using ISSM's built in bidemnsional anisotropy mesh generator, we refine the mesh using a metric based off the product of bedrock slope and surface velocity (strain rates). The range of individual element edge length ranges from 400m-4km (high-low strain rates). We will expand upon the methods section to include this information.

Line 82 – 85: 'A linear-viscous sliding relation…' How does the inversion arrive at appropriate coefficients for the sliding law? I don't understand what is meant by 'which have been scaled down to give a smooth transition between both data sets', please expand on this.

We will expand on the process for how we determine sliding coefficients for the Budd-Glenn sliding law in the methods. We first patch 2 surfaces of 1985 and 2009 and assume ds/dt is proportional to velocity as to offset the 2009 velocities. We then composite the 1985 and 2009 velocity series. We will clarify this information in the methods.

Line 86-87: What are the boundary conditions elsewhere in the domain?

At the catchment boundaries, we impose ice temperatures from Seroussi et al., [2013]. Along the computational domain's lateral boundary, observed ice flow velocities are imposed and water pressure at the calving front. We will clarify this information in the methods.

Line 95: Can you explain the correction that has been applied in more detail? I don't understand what has been done from this sentence.

We utilize a surface elevation model of Sermeq Kujalleq's lower elevation area from 1985, derived from photogrammetry [Korsgaard et al., 2016]. We address data gaps inland by incorporating surface elevation data from the Greenland Mapping Project [Howat et al., 2014], adjusting this data to match the earlier dataset with an elevation offset that corresponds to current ice flow velocities [Rignot and Mouginot, 2012]. We will clarify this information in the methods.

Line 112: Is the linear decrease in calving stress threshold supported by previous studies, observations or physics?

The linear variation in calving stress threshold is a simplified realization of the effects of mélange weakening/strengthening on buttressing the calving front, which has been posited as a possible explanation for observations indicating mélange weakening and breakup in concert with ocean temperature seasonality, rather than atmospheric temperatures [Kehrl et al. 2017, Bevan et al. 2019, Joughin et al. 2020]. Our main justification for this assumption of ice mélange

weakening in response to heightened temperatures and we use a linear sensitivity a simplest case option and for easy comparison to other linear sensitivities assumed in this study.

Line 114: Equation 3 – explain what sigma_max and sigma_min used in this equation are.

Sigma_max represents the maximum stress threshold corresponding to the minimum temperature within our temperature time series. Sigma_min representns th minimum stress threshold corresponding to the max ocean temperature. When ocean temperatures are warmest, we assume mélange is at its weakest and therefore proclivity for calving is intensified and vice versa. We will clarify and standardize usage of parameters in methods.

Line 129: Equation 4 – A and B are not explained. Are these tuning parameters? What are their values?

A is a tuning parameter with a value of 3E-4 (non-dim) and B is to ensure that heat flux does not vanish in the absence of melt water and has a value of 0.15 [Rignot et al., 2016.] We will clarify this information in text.

Line 139: Equation 5 – Add a table with values of parameters such as gamma_T etc.

We will add a table with variables and their respective values at the end of the methods section.

Lines 144-150: How the parameter space is varied may be better suited in the respective sections 2.2 and 2.3 rather than in the section detailed the mismatch score.

We will move this section regarding how we vary parameter space earlier to sections 2.2 and 2.3.

Line 145: 'we choose a range of 0-4x the empirical parametrisations' What are the choices of range based on? Are there any observations / literature to support this?

There are no literature based reason for this. This range was selected following a series of experiments within different ranges to capture as much variability in simulation output. We do this instead to capture various forms of interplay between ocean-temp dependent calving and submarine and frontal melting. We will elaborate on this more in the methods.

Line 146: 220-360kPa – same comment as above. Please justify the choice of ranges. Figure 2 shows results up to 350kPa, should this be 360kPa?

Results are only shown up to 350kPa. Beyond 360 kPa, the calving front and grounding line do not retreat at all and did not yield valuable information. We will include this information in the methods.

Figure 1: It would be helpful to know where the grounding line is and how this moves through the simulation. A lot of the results interpretation talks about the ice tongue and it would be very helpful to visualise how the extent of the ice tongue changes and a note to emphasise when it disintegrates.

We will include a supplemental figure showing the grounding line and calving front migrations for our best performing model.

Figure 1: Is 'Elevation' shown colour map 'Bed elevation'?

Yes, elevation refers to bed elevation. We will correct this in the figure.

Figure 1: I find the figure somewhat difficult to interpret. As the modelled and observed calving fronts deviate quite widely it is difficult to tell which dashed lines should be compared to which solid lines from the colour scheme. Is it necessary to present data from every year or can the colour scheme be adjusted to make it more clear which modelled and observed calving fronts should be compared?

We believe that presenting data from each year highlights the differences in rates of retreat between periods of rapid and slow retreat. As such we include one front per year. We have experimented with plotting fronts more sparsely or labeling individual contours, but these missed presenting years in which critical retreat occurred or cluttered the figure so that it was difficult to read.

Line 154: No need to write 'within' as well as 'less than'

Fixed in text

Line 155-162: I don't understand the scoring system from the description given here. Please can this be addressed and made more explicit?

We added in text, for each observational data point, we calculate the difference in surface area between observed and simulated geometries. We then take the RMS of the vector of differences to derive an overall mismatch "score". We will clarify this in the methods.

Line 161: How is the error vector (starting and end point) defined?

Refer above. The first timepoint corresponds to the mismatch at t1(1985) of the simulation and final timepoint corresponds to tend(2018). We will clarify this in the methods.

Line 168: 'thin lines' should be dashed lines

Corrected in text

Line 169: 'thick lines' should be solid lines

Corrected in text

Line 177: For discussion on the disintegration of the ice tongue, no grounding line locations are shown in figure 1 so the location / extent of the ice tongue is not clear

Noted in text that grounding line is located at a specific distance upstream from calving front. We will also provide a visual reference of grounding line migration as a supplemental figure.

Line 186-187: It would be interesting to see the evolution of tensile stress seen at the terminus of both branches over the years. Can a figure be used to illustrate this?

Added figure to supplemental info showing glacier stress at key times.

Figure 2. It is not clear why the specific parameters are presented in panels b – d.

Corrected in text to standardize symbols for parameters and explain meanings.

Figure 2: Is M_f the same as M_fr (given in equation 4)?, Is M_s the same as M_sm (given in equation 5)?

Corrected in text to standardize symbols for parameters and explain meanings.

Figure 2: What is the red dot displayed on panel b? Best overall fit? Please add explanation to caption.

Corrected in text to indicate best fit model.

Figure2: alpha_ms and alpha_mf are not defined anywhere, are these the same as M_s and M_f?

Corrected in text to standardize symbols for parameters and explain meanings.

Line 198-199: '…than suggested by the two-equation parameterisation.' Add citation for this.

We will add a reference here [Rignot et al., 2013]

Line 204: With the best fitting simulation requiring high sensitivity to frontal melting, how do the ablation rates m_fr and c compare in the parameter space? It would be interesting to see whether the calving rates and frontal melt rates vary on the same magnitude.

This is a great suggestion and we thank the reviewer for highlighting it. We will include a supplemental figure showing the following, a single panel plot similar to 3a, but for all 3 parameters and for every simulation in the parameter space where line color would correspond to RMSE. For example, panel a) calving panel b) frontal melt and panel c) submarine melt sharing the same legend

Figure 3: Panel B. Observed area change time series vs modelled area time series would be more informative.

We will correct in figure 3.

Line 218: 'SK maintained a floating ice tongue ahead of its terminus'. Is the terminus not the front of the ice tongue at this point?

SK maintained a floating tongue which at some locations extended upwards of 10km ahead of its grounding line. We will clarify this in text.

Line 239: Need figures to support this statement

We will include a reference to figure 3a.

Line 230: Calving rate in this study is defined by speed, stress and stress threshold. Temperature can only play a role through these variables, why would we expect calving fluxes to be controlled by temperature? The calculation of 'calving flux' should also be defined as different definitions appear in the community (not as unique as 'grounding line flux')

We will clarify in text the units we use for our calving flux. We expect that, in our simulations, our calving fluxes would increase as temperature increases due to the temperature-calving stress threshold dependence we impose. We note here however that there is a lag between the initial temperature increase in'96 and the subsequent increase in calving flux in 2010. We will clarify this information in text.

Line 269: With the runs up to 2100, significant terminus retreat is observed. Is the same mesh considered as had been described previously? Is the mesh resolution at these retreated grounding line locations still 400m?

Mesh resolution scales from 400m at the initial front to 4km inland. While the mesh is certainly coarser than 400m by the time it retreats far upstream, the short time step, enforcement of CFL in the model and use of a level set to define the terminus position ensures that the model is still doing a good job representing terminus fluxes. We will clarify this information in the methods.

Line 272: How is the range of threshold values 260-437kPa chosen? Same comment as on Line 146.

We will discuss in supplemental info why we chose range of 260-437kPa.

Line 278-279: 'even if ocean temperatures returned to the coldest values achieved during the historical period over the next 80 years, the rapid acceleration of calving and retreat would likely continue unabated' – this may well be due to a limitation with the calving law which should be acknowledged, rather than necessarily being a direct indication of what may happen in the future.

This is a good point, and pretty much what we were trying to get across with these simulations. In these simulations, even when ocean temperatures returned to their 1985 values, we see that SK continues to retreat. However, our interpretation of this result is not necessarily that this implies SK is in the midst of an irreversible calving-driven retreat in reality, but rather that simulations which represent calving in such a simplistic way will tend to produce irreversible

retreat, but this may not be the case if mélange is realistically represented. We will clarify this information further in text.

Figure 4: caption – What is the 2100 run? The ensemble members running from 1985 – 2100? The caption could be more concise.

We will clarify what our 2100 runs correspond to more clearly in text. We will include a new caption: Calving fluxes of 1985-2100 run ensemble members. Between 2018-2100, maximum calving stress threshold is set to a multiple of the calving stress threshold at 2018.

Line 305: sigma_max or sigma_thr?

We will correct to SIGMA_MAXthr

Line 315-320: See 'Parsons et al, 2024, Quantifying the Buttressing Contribution of Sea Ice to Crane Glacier' for proposed methodology for assessing the terminus stress regime with/without adjoining melange elements. Could a similar method be used in this study to account for changes in the terminus stress regime with the presence of melange?

We will cite Parsons et al., 2024 as a potential way of quantifying buttressing impact in text.

Figure 5: Panel A. Improve the quality of this figure. It appears to have been stretched.

We will improve figure 5's quality.

---

## Author Comment (AC2)

We would like to thank the reviewer for their insightful remarks. We note for the reviewer and the editor that there are many new additions and insights provided by this study, as compared to prior studies, of which the reviewer makes note of several. We found the reviewer's comments to help us identify places where we can further highlight the new insights provided by our novel experimental design, and that the revised paper (which is already mostly completed) will be satisfactory to address the reviewer's concerns. Below is our initial response, broken down by remark original review comments are shown in black, while our answers are shown in red.

*Novelty

I think the main new addition in this paper is the parameterization of ice melange effects. The authors do that by making calving stress threshold a linear function of ocean temperature. It is not clear how significant this addition is, as there is no comparison to simulations without this feature. Previous studies using the same model (Bondzio et al) were able to reproduce SK evolution just as well, so it seems that the proposed form of melange parameterization is not necessary for reproducing the past. This is a major issue for the claims and findings in this paper with regard to the importance of melange to SK dynamics - essentially there is no control run presented here to evaluate this importance against.

First off, we thank the reviewer for showing how our framing of this study was insufficiently clear so as to produce some misconceptions about our aims. To be clear: our aim throughout this study was to explore the drivers of SK's retreat during the period of 1985-2018. Additionally, we show that we can capture SK's evolution well without forcing the front with observed velocities as in Bondzio et al. [2017]. Our aim was not to accurately simulate mélange (which is not currently possible with current modeling systems, though the efforts are underway from the authors to remedy this), but to offer a scenario where disentangling SK's retreat can be accomplished with the considering of long-term changes in mélange buttressing of the terminus through modification of the calving threshold parameter. We place our results in context of Bondzio et al. [2018], thus making those prior simulations akin to a "control" that we compare against. Importantly, by including interannual variability in the calving threshold parameter driven by ocean forcing, we do not require 4x multipliers of frontal melt, as Bondzio et al. [2018] did to achieve reasonable match between observations and simulations. Our aim was to present a different perspective on how ocean-temperature-based processes could modulate calving in the context of SK. Given the central importance of SK to discussion and parameterizations of climate forcing of dynamic glacier retreats (i.e., calibrating ice cliff wastage parameterizations), our results provide an important advance showing the central importance of interannual changes in calving to explaining SK's observed retreat without resorting to unphysically high frontal melt rates. We strongly believe that by clarifying these aims, the reviewer's concerns about the importance of this work should be addressed. We have significantly re-written the introduction to highlight these aims more clearly.

*Goals and Scope

*The paper needs to better clarify the questions it is addressing and how the answers it comes up with support, or contradict what is already known about the SK retreat.

Our main question concerns the rapid retreat of SK and how melt and calving processes could have driven its evolution from 1985-2018. Beginning in 1998, the rapid retreat SK's calving front initiated a runaway mechanism wherein mélange could have played a large role in modulating calving rate. We produced a large-ensemble where we could compare relative contributions of melt and calving processes in the presence of weak and strongly sensitive mélanges. We mainly note that we do not need unphysically high melt rates to recreate SK's evolution but that beyond the period of retreat we notice a rapid runaway effect, suggesting that the calving dynamics which would accurately describe 1985-2011 evolution may not be suitable for later periods. We will clarify on these questions we are addressing in text.

There are multiple papers by Bondzio et al that concern SK, and I believe the authors here use the same model and try to build on this work. One of the papers of Bondzio focuses on disentangling drivers and mechanism of the retreat. How does your study compliment this work. Or other existing hypothesis of the retreat?

Bondzio et al. [2017] aims to explain SK's evolution from 1985-2015 using a 3-D thermomechanical model configuration. In that study the evolution of SK is forced using observed calving front positions and the authors find that "the viscosity drop, the trough's low basal drag, and the geometrical adjustment are the main drivers behind the observed inland acceleration of JI". Following up on that, Bondzio et al. [2018] to identify the ocean's role in SK's present and future mass loss. In that study the authors find that a seasonal calving cycle and ocean-temperature forced melting accurately capture's SK's behavior. Specifically, Bondzio et al. (2018), use a dynamic seasonal calving cycle which switches calving thresholds between 4MPa in the winter and a prescribed summer stress threshold which is varied from 60-180kPa. They find that models with summer stress threshold of 150kPa most accurately captures observed behavior, with the caveat of frontal and submarine melting having to be amplified to 4x and 2x respectively. Since Bondzio's studies came out, substantial importance has been placed on the role of calving in describing the very fast retreat at SK, and whether the ocean has a role in explaining changes in calving (e.g., Joughin et al. 2020). Therefore, there was a need to further study whether interannual variations in the propensity for calving (through sigma_max) explain (even partly) why calving has increased quite so much as SK. This is where our study comes in: we follow up on this work by addressing whether we can capture SK's behavior without employing such large melt amplifications and find that we can do so by allowing ocean-temperature calving dependence. We will contextualize our study more clearly in the aforementioned context and include a relevant section in methods.

The authors write "there is still debate about which physical processes are responsible for mediating the glacier response to ocean warming" but the reader never learns about what the debate is and how this study contributes to this debate.

We will highlight and clarify in the text the following hypotheses

(1) Processes occurring at tidewater glacier termini act to trigger a retreat and reduce buttressing stress. These processes are most commonly calving and mélange buttressing. Calving is further amplified by basal over deepening and retrograde sloping beds while mélange is primarily generated via calving and experiences a seasonal variability in response to fluctuating sea and air temperatures.
(2) The second hypothesis suggests that enhanced surface melting in response to warmer air and ocean temperatures initiated ice tongue collapse and subsequent inland ice velocity acceleration.

How about the papers of Nick et al. and similar, they seem to be relevant to this paper.

Nick et al. (2009) report that Greenland's outlet glaciers "adjust extremely rapidly to changing boundary conditions at the calving terminus." Additionally Bondzio et al. (2017) further affirms calving-front dominated evolution in the context of SK. This reinforces our findings in that we recognize that the calving front dynamics are a much greater control on retreat than submarine melt processes. We will include mentions of these studies in text in addition to a clearer contextualization of our study.

For an example how to clearly state hypothesis and place your study in existing context, see for example Bondzio et al. 2017: "The mechanisms behind Jakobshavn Isbræ's acceleration and mass loss: A 3-D thermomechanical model study"

We will re-introduce the hypothesis following the points made above.

*Methods

There is no information about the temperatures used to prescribe melting. A mention is made about ECCO, but no detail what so ever about how are these temperatures propagated/extrapolated into the fjord, whether topography, namely the sill, is taken into account during this extrapolation. Some comparison of existing vertical temperature profiles with whatever vertical temperature profiles the authors come up with would be good to show too.

We simulate the process of submarine melting by assuming that the water column is stratified such that water at maximum depth is also the warmest water. We use this assumption to justify the warmest waters overcoming the shallow sill. Following this, we take the depth-average of the water column's temperature between the surface and maximum of bedrock and sill height (-250 m). We fill the data gap between 1986-1992 by repeating the temperature time series between 1992-1996. We will incorporate this information in the submarine melt section of our methods.

Clarify when is frontal melt prescribed. Only when there is no floating tongue? Or also on the vertical calving front of the floating tongue? If the latter is the case, then how do you account for the effect of convective forcing due to meltwater from basal melting emerging at the front - do you have an estimate for that?

We do not estimate or account for the effect of convective forcing. Frontal melt is applied to the front of the glacier and only to submerged areas. Submarine melt is applied to all ice below sea-level on the ice-ocean boundary. We do not account for convective forcing due to meltwater, because the sensitivity to subglacial discharge is lower than to ocean temperature (as in Xu et al. 2013). We will further highlight this fact in the methods.

There is no information about which parameters are varied throughout the sensitivity studies. These parameters then appear, unexplained, in figures, but they are nowhere introduced.

While the previous version does provide an explanation of the various parameters, there was some inconsistency in symbols and explanation across the text. We will fix this by standardizing symbols across text and figures and summarizing the parameters perturbed in the ensemble at a single point in the text.

*Validation

There are some strange unphysically looking calving front positions that may indicate problems with the calving algorithm. Specifically, I am referring to the appearance of two structural arches, well before the glacier splits from one through into two (I would give you a location but there are no axes on the plot). Can you explain this?

Yes we agree that remnant ice is an artifact of our calving algorithm. However, the features that the reviewer has identified here are transient and we thank you for pointing it out. However, these are transient features which do not appear in all simulations. Even in simulations with the some overall retreat history, few have these features and others do not, and so we discount their overall impact on the main results of this study.

None of the simulations appears to be able to stabilize and readvance once it has begun to retreat, unlike what happened in observations around the 2017 cooling. This indicates that the model is more eager to retreat than in reality. This needs to be addressed.

The reviewer makes a good point here, we refer you to our discussion on lines 336-345 in particular we note that our simplistic calving tends to prioritize retreat when we select for the lowest RMSE model. This is due to the observed rapid retreat of the calving front in the 2000's which favors models with parameter combinations that allow for a rapid retreat during this period. However, we miss an important negative feedback on calving in the form of mélange which would inhibit potentially slow or fully arrest model runaway retreat as has been previously identified by Khazendar et al. (2019) and Joughin et al. (2020) as an important factor in describing the readvance in 2017. We will further highlight this shortcoming in the discussions section.

*Results

The results are not in general supported by figures. There needs to be a clear separation between results and discussion/interpretation and 'general knowledge' statements which assume that the reader is inclined to agree with the authors. All results should have figure references, which is currently almost never the case. This is a major issue in assessing the manuscript contribution.

We thank the reviewer in pointing this out and we will address the lack of in-text figure references by providing relevant references to claims we make in the discussion and results section.

There is no stress quantification that would support several claims and arguments by the authors.

We will include a contour plot of stress field for the best fit model with snapshots at relevant points in time: Initial, Disintegration of ice tongue, Initial calving acceleration and Final times.

The results seem to suggest that the main ice tongue disintegration occurred via basal melting and not by calving (Fig 3a). Is there observational evidence for that? I was under the impression that there were some major calving events that diminished the floating tongue, rather than in just melting off.

While submarine melt was the primary source of mass flux during the disintegration of the ice tongue, we do not want to imply that as the sole reason for ice tongue collapse in our model. Observational evidence shows that a series of calving events and consequential front retreat occurred during periods of looser ice mélange and heightened water temperatures in waters proximal to where the fjord enters Disko bay. Hypothesis concerning ice tongue collapse focus on 1)hydrofracturing induced by surface melt filling crevasses and 2)ice mélange weakening influencing SK's terminus stress state by reducing frontal buttressing and consequently facilitating calving activity. We will include these hypotheses for ice tongue collapse to further contextualize and reinforce our assumption in SK's evolution being largely driven by frontal dynamics.

*Hysteresis

The authors run experiments with different calving stress thresholds into the future(? unclear what temperature forcing they are using). And then claim the system experiences hysteresis, but they don't actually do experiments with reversal of the forcing conditions. While it seems perfectly plausible the system experiences hysteresis, it was not actually shown in this paper, and as such it remains a hypothesis.

We thank the reviewer in pointing this out and we will elaborate on our methodology. We conduct our hysteresis experiments by taking the best fit model and holding temperature forcing (cold ocean conditions producing low frontal melting) at its 2018 configuration. Since we assume that calving is the dominant control on SK's evolution by 2018, we vary the 2018 stress threshold from 1-10x for a total of 20 models. We term these "2100 runs" and were interested in whether we could see simulated SK readvance under these conditions. Thus, we do conduct a hysteresis experiment in that we reverse the calving parameter to much higher values, including those previously attained when calving rate was low, and even above those values. The point is that the evolution of the system during the period of low sigma_max causes it to enter a state that is not reverse when sigma_max is returned to these values or even above these values. This fits the definition of hysteresis (e.g., see the classic textbook on dynamical systems by Strogatz, p. 60). We will include this information in the results and further discuss their implications and shortcomings in the discussion section.

*Discussion/conclusion

There are some relatively strong statements at the end of the discussion and conclusion sections advocating for a fully dynamic and coupled model of melange. I don't see how any of this need is supported by this paper. What was done here is only a very simplistic parameterization of calving rates on ocean temperature, and as admitted, no feedbacks between melange strength and calving rate were incorporated. There is probably still room for more sophisticated parameterizations (e.g. following Schlemm et al.)

We thank the reviewer for pointing this out. We will add further discussion of mélange parameterizations such as that of Schlemm et al., which should help acknowledge some of the deficiencies that come from not representing mélange in ice sheet models. However, such simple parameterizations are ultimately meant as upper bounds since they do not represent the true rheology of mélange. New approaches using ideas from granular physics (i.e., Amundson et al., Currently Accepted at TC) to dynamically model mélange are necessary to accurately predict the buttressing force of mélange, and its interactions with the ocean and glaciers. We ultimately wish to use the results of this study as a motivator for why we must develop more robust melange models which would hopefully be integrated into regional ice sheet models. Especially since we note that terminus dynamics impose a significant control on overall glacier dynamics, it is imperative that we develop a proper representation of the seasonal and calving-associated mélange dynamics. We will flesh out the discussion of mélange parameterizations and models in more detail in the text in response to this suggestion.

IN LINE REVISIONS

INTRODUCTION

"I don't see what you mean by this and how that has anything to do with retreat"

Here we are referring to the ice tongue's ability to buffer melting near the grounding line by acting as a heat sink for warmer fjord waters. We will clarify this in text.

see Gladish et al. 2015: Oceanic Boundary Conditions for Jakobshavn Glacier. (Part I and II but mainly II) The oceanographic analysis there shows that the suggestion of Holland at el 2008 about warm waters actually making it to the Illulisat Fjord is actually quite simplistic and although widely accepted it has a lot of caveats.

We recognize that while warm water may not have directly reached SK's front, we keep consistent in our assumption of intrusion as to compare to Bondzio's study. But we will include references to highlight the possibility that warmer waters may not have intruded as deep as assume here and make explicit mention of this in-text.

what is the debate and which are the relevant papers on that?

We aim to identify whether SK's observed retreat can be attributed to an amplification of melt due to increased water temperatures or a reduction in buttressing due to a weakening of pro-glacial mélange leading to amplified calving [Thomas et al., 2004]. We refer the reviewer to our comments made earlier in the overview regarding our hypotheses and the relevant debate. We will clarify these hypotheses in the introduction and are thankful for the reviewer's input.

during which timeperiods did those occur? Is it based on direct temperature measurements in Disco Bay as are the data from the more recent time periods?

We will remove the sentence.

Do you model this period though? When does this warming occur? I don't see any warming in Figure 3c.

We will correct this sentence to: "why did the most recent period of warming beginning in 1985 cause such a dramatic and unprecedented retreat?"

But there are also counter examples where these aren't at all related, see Amundson et al., 2020, Le Conte Glacier

While counter examples do exist, we note that here our analysis is strictly contained to SK. However, we will make explicit references to these cases and hypothesize why it might be more important in the case of SK (melange and frontal melt) than Le Conte.

Maybe cite Xie et al for SK melange dynamics observations?

We will add a reference to Xie et al., 2019

what is direct melting, as opposed to indirect?

We will fix to "Disentangling the drivers behind SK's response to warming ocean conditions requires distinguishing between retreat driven through **ocean-induced melt** and calving."

why are icebergs an issue? which observations do you have in mind that would solve the problem?

Observations of environmental variables such as mélange density, detailed calving catalogues, geometry of calving front. Icebergs act as physical obstacles which may complicate data collection methods, especially near the glacier front. We will clarify why these observations are relevant and their associated difficulties more clearly in text.

calving rate? or threshold?

We will fix to: We perform a large ensemble of simulations of SK retreat through perturbation of three sensitivity parameters that impact its retreat: subshelf melt, melt at the calving front, and **calving threshold** modulated by mélange rigidity.

METHODS

The variability of which quantities are you actually comparing? If SMB and temperatures then that is not meaningful - perhaps 'small' SMB variation can produce an equal effect on submarine melt as 'big' ocean temperature variation.

Here we are pointing out that the relative variations in SMB are nominal relative to the relative variations in ocean temperature. We make this assumption as we are interested in disentangling ocean-driven mechanisms. We will clarify this in text.

which ones specifically?

Here we are referring to surface mass balance forcing. We will clarify this in text.

where? local? remote? This needs some elaboration.

Here we are referring to local surface mass balance forcing. We will clarify this in text.

more than what?

We will fix to: "…be more influential in it's role modulating ocean conditions **than it is in effecting SK's evolution via direct variations in SMB**."

can you give examples of these processes?

We will fix to: "The stress threshold parameter can be thought to conceptually represent many physical processes **such as fracture toughness, grain-scale deformation, and ice strength**, which have the ability to modify the propensity for calving events."

explain each symbol in text

We will add in-text explanations for each symbol.

this is sigma_min? or min(sigma_thr)?

This is the minimum stress threshold which corresponds to the highest temperature in the respective disko-bay temperature time-series. We will clarify this in text.

what is the max value?

The max value is the parameter we vary, in effect we are varying the calving threshold's ssensitivity to ocean temperature. We will clarify this in text.

But when there is floating ice tongue, part of the basal melt water that emerges at the calving front in a way acts as a subglacial discharge in that it produces a convectively forced plume rather than an unforced one. So that may need to be accounted for?

We note that while this is a potential short-fall of the frontal melt parameterization, beyond the removal of the ice-shelf, we would not need be concerned with convective plume forcing, at least within the context of our study. However, we will make a note of our omission of plume dynamics in our methods and are appreciate of the reviewer's point.

Do you mean basal, as opposed to frontal?

Submarine melt acts on floating and grounded ice exposed to ocean water (below sea level). We do not include basal melting processes in our simulations and will clarify this in the methods.

what do you use for gamma t? presumably that is a function of friction velocity - do you assume constant? I think this paper needs a table with symbols, their meaning, and values for constants (and references where they are taken from if that applies)

We assume the specific heat capacity of the mixed layer is constant and use the same value as in Bondzio's relevant work. Additionally, we will add a table with relevant parameters and their values.

how do you get this value? Is that just directly take from ECCO?

We use a constant melting point value of -1.85 C as an averaged value obtained from the equation of state with pressure and salinity values obtained from ECCO. We justify using a constant melting point value in that we expect its variation to be an order of magnitude lower than variations in ocean temperature. We will clarify this in text.

I think the methods section could use a summary of what are the three parameters that you are actually varying. Or state it somewhere else, but should be really clear and explicit.

We added a table with relevant parameters and their values. We also clarified the three parameters which we vary with a summary sentence at the end of the methods section.

what does this mean? what are you actually varying in the parameterizations? some factor that is not included in those equations? If that is the case, please include it

We vary the magnitude of melt obtained through empirical parameterizations through a simple multiplier parameter. For example, our parameter space for submarine melt would include 0x, 0.5x, 1x, 1.5x,… the base parameterization for submarine melt. We will clarify the distinction between a parameterization of a process and multipliers we use in our parameter space in text.

put axes on plot, or coordintates

We will add axes to the plot.

Can you mark when in observations and when in simulations  the glacier looses the floating tongue? It is not clear from the calving front positions, as I assume those are either floating or grounded.

We will add in text reference to grounding line location when SK loses its ice tongue. Additionally, we will include a supplemental figure showing grounding line locations for our best fit model.

That is not quite true. You want to be able to capture quiescent times as accurately as retreat times.

We note that our aim is to capture quiescent as well as retreat periods, we are just highlighting that weighing observations based off observation density would not affect our analysis due to periods of rapid retreat coinciding with the highest observational density. We will expand our reasoning as stated here in text.

RESULTS

I am not sure what this whole segment is - is this results? If yes, then there need to be figure references associated with each statement, and supporting it. Is it introduction? - in that case the statements need to be referenced. Is it discussion of results? In which case it should be placed after the results that are supported by figures.

We thank the reviewer for pointing this out and we will Include specific figure reference to support our results.

I don't see any 'pause' in the simulated extent once the glacier starts retreating (yes it hangs on a bit at the initial position)

We will clarify in text that behavior of arrest at pinning points is behavior witnessed across ensemble members and not specifically only within our best fit model.

Except for the abilty to readvance

We discuss the issue in capturing re-advance later in the discussion.

why don't you just include it in the plot?

We will update the plot to include the entire ensemble

Can you explain which part of the figure and what region you are referring to?

We will clarify in the text that we are referring to the area of the parameter space within a high maximum stress threshold regime and a lower submarine melt multiplier.

what is that - how do you define it (the cold ocean part), and where did you conclude this from? which portion of which figure?

We will reference Figure 2a) in text, and re-clarify earlier and within this section that the maximum stress threshold parameter is equivalent to the stress threshold when the temperature is at its lowest within the time-series of ocean temperatures.

What is the red dot in panel b?

We will clarify in Figure 2 caption. The red dot indicates the model parameter combination which achieved the lowest RMSE.

Is $M_s$ and $M_f$ the same thing as $m_{sm}$ and $m_{fr}$ in text? Can you just it make it self consistent?

We apologize for the confusion and will clarify and standardize symbol-parameter combinations through the text.

explain legend what is alpha and other symbols used in the figure

We will fix the legend to indicate alpha as the multiplier to our parameters.

which parameter are you changing?

Here we are not changing a parameterization of melt, but we are referring to increasing the submarine melt multiplier. We will clarify this in text and will keep consistent in parameter/multiplier usage in text.

Are you actually testing different basal melt rate parameterizations? If not, how did you conclude this? If yes, which ones?

Here we are not testing different basal melt conditions instead we are making the point that a submarine melt multiplier <1 is required for models to achieve a lower RMSE. We will clarify this in text.

Again, point to figure

We will reference Figure 2 here and point the reader towards subpanels specifically to note relationships between parameters.

to compensate what?

We will change in text to: "…such that simulations with higher cold-ocean stress thresholds (i.e., less calving in coldwaters) also need higher submarine melt rates **to achieve reasonably low RMSE**"

You haven't define what Mf is and how it enters which equation

We will define what Mf is earlier and keep consistent through text.

I am a bit intrigued by this plot. Are you saying that calving did not at all contribute to the SK retreat? so the changes in the calving positions during that time are just a result of the frontal part of the ice tongue melted off? How does that compare with satellite observations?

We will include supplemental info plot to show initial stress state and as calving picks up as evidence for why calving fluxes become significant towards the beginning of 2010. We agree that calving resulted in the loss of the ice tongue and we highlight later the shortcomings of our simple parameterization of a temperature-based calving threshold. Specifically, since we are interested in the model with the lowest RMSE, we require a larger stress to accommodate later periods of retreat which may make the initial breakup of the ice tongue less dependent on large calving events as noted in observations.

Something like a plot of ice shelf basal area time series would be helpful here - to determine when the ice tongue is lost and why different terms gain importance at different times. And also timeseries of ice front area for the same reason.

We will include relevant figures of ice shelf basal and frontal area in the supplemental information.

simulation?

We will correct in text to "Best fitting model"

in observations or in simulation?

We will clarify in text that this paragraph is referring to observed ice tongue behavior not simulated behavior.

why? is there an evidence of seasonal cycle in temperatures inside the illulisat icefjord? I think you claim that subglacial discharge is not important elsewhere in the manuscript, but if it induces a significant seasonal cycle in melt rates, can you maybe clarify/remind the reader here?

We make this claim with respect to the observed behavior of SK's ice tongue, before its collapse. However, we thank the reviewer in pointing out our omission of calving as a component of balance during the period of 1985-2000. We will include further clarification highlighting that ocean melt AND calving were in balance with incoming upstream ice flow and surface accumulation pre-collapse.

that is a result from simulations? of that is interpretation of observations?

This is an interpretation of observations. We will clarify this in text.

I dont understand what you mean by this and how that buffers retreat

Here we are referring to the ice tongue's ability to buffer inland ice from warmer surface ocean temperatures but not necessarily melt at deeper depths. We will clarify this in text.

Where, local in the ice fjord? or local further on the shelf?

Local to the ice fjord. We will clarify this in text.

Add some measure of that into the time series?

We will include a supplemental figure indicating stress state at key points in best fit simulation.

ok, here it would be good to have some time series indicating the ice tongue base steepening

We will include a supplemental figure indication glacier profile at key points throughout our simulation.

which part? the one with 1 branch or the one with 2 branches?

We will change in text to "southern branch".

Same comment as before, this whole section needs figure referencing

We will include more in-text figure references to evidence claims.

how did you determine that

We will reference Figure 2 and note that the by 2010, SK has retreated into a much deeper and narrower bed topography.

which type of melt?

We will fix in text to submarine melt

Add point on figure indication loss of ice tongue

We will mark Figure 3 to indicate time point of ice tongue collapse.

Horizontal only, correct? You don't take into account vertical melting profile and its role in calving, right? Can you make that explicit somewhere

We will clarify earlier in the methods that we don't take into account a vertical melting profile and that melt acts horizontally.

cite them

We will include citations to studies highlighting potential hysteresis at SK.

We will fix the typo in referencing Kajanto et al. (2020)

what about frontal melting?

We will include in text, "driven by calving fluxes **and frontal melting** as floating ice area considerably decreases and the effects of submarine melting are diminished."

but that doesn't seem to be so consistent with observations, in which the ice front position is somewhat stabilized, definitely in the northern trunk

This is not consistent with observations and highlights the lack of a necessary negative feedback to address runaway calving within our model. We will clarify this inconsistency in text.

cite

We will include citations referencing high mélange output due to rapid calving

How do you conclude this, without actually doing tests with temperature reversal?

We will make a point in text to highlight that we are not reversing temperatures, instead we are setting ocean temperatures fixed to their 2018 state and focus our hysteresis analysis purely on increasing calving stress threshold. We refer the reviewer to our earlier response addressing hysteresis in the reviewer's overview.

DISCUSSION

which ones?

We will fix in text to "…that the Budd(1979) basal sliding law and coefficients captured SK's behavior fairly well. And include a reference to Budd et al., 1979

These were never introduced

We will introduce and define and keep consistent parameter symbols.

multiple factors larger than observed.

We will fix in text to "…fronts which closely match observed calving fronts without requiring melt forcing to be multiple factors larger than suggested by our empirical parameterization."

the only way for your model and for these particular parameterizations

We will fix in text to "The extended simulations described in section 3.3 indicate that the only way **for our best fit model** to arrest the future retreat of SK…"

where did you show that?

We recognize that we do not show this and we will remove the following text to "The extended simulations described in section 3.3 indicate that the only way to arrest the future retreat of SK is by increasing the calving threshold to values never attained during the historical period in the best-fitting simulation. ."

While this would be great, I don't think having a dynamic model of the melange, is not any more necessary than having a dynamic model of the ocean. You seem to be comfortable with a simplistic parameterization of the ice-ocean interactions, but these may be just as important (or more) to simulate, than the melange dynamics.

We thank the reviewer for making this point and agree that while our parameterizations are simplistic in nature, they still highlight that without some form of dynamic calving in response to ocean conditions we would need much higher melt rates than suggested in order to achieve a reasonable match to observations. This is compounded by the fact that had warmer waters not intruded into Disko Bay as far as assumed then local fjord conditions would represent cooler waters and require even higher melting rates than assumed by Bondzio (2018). We thus use these reasons to assess that a dynamic mélange component would provide the greatest constraint on the evolution of SK compared to other ice-ocean processes. We will further clarify this in text.

Fixed typo "and very computationally expens**ive** tools"

**CONCLUSIONS**

Do you actually use vertical temperature profile from ECCO in Disco Bay proxy, or do you just use a single temperature point? Do you take into account the sill discussed in Gladish et al? The first time we here about deep water in the manuscript is in conclusion!

We take into account the sill when building our temperature forcing time series. We limit temperature forcing to depths above the sill and take the average of the water column as a single point. We will include this in text.

Not sure what you are referirng to here and why it is important and what is the reason for that.

We will move this sentence to the end of paragraph 2 in conclusions to highlight the error arising from our simplistic ocean temperature based calving.

All this should have appeared in discussion at some point and been elaborated on there.

We will move these sentences to the hysteresis section of our discussion.

Can you mark that bimodality on figure 2 in some way?

We will include figure showing a histogram of RMSE along our flowline in the supplemental information.

do you mean metastable?

We thank the reviewer for the suggestion but believe quasi-stable and metastable are interchangeable in this context.

I don't think you have shown the likelihood of this in any way. It is just your hypothesis. There are a lot of simplistic parameterizations in your set up, so any of those is a potential candidate.

We will clarify in text that our ocean temperature based calving parameterization simplicity is the most likely culprit of runaway retreat.

This is not at all what was shown in this paper, as far as I can tell.

We will further clarify upon this conclusion by first reaffirming that our simplistic parameterization of ocean temperature-based calving is a first order approximation of mélanges potential effects on calving. We will also recontextualize the importance of frontal dynamics specifically on the evolution of SK.

This was not shown anywhere in the paper

We will include a supplemental figure of the stress state to back this claim.

This is shown here? Or that is hypothesis about what happened in reality?

We will note in text that this is our hypothesis about what happened in reality.

---

## Referee Report (RR1)

**Originality**

This study builds upon existing literature whereby frontal and submarine melting rates and calving rates were optimised through ensemble modelling to reproduce the observed retreat of Sermeq Kujalleq. This study employs a new calving rate parameterisation whereby the threshold stress which ultimately controls the calving rate changes linearly as a function of temperature. This is proposed as a simple modification to a tensile stress based calving law which accounts for seasonal and interannual variability of ice melange and its associated buttressing potential.

**Scientific quality**

The aims of the study are well defined and have benefitted from the manuscript update. The methodologies are well defined and the assumptions and limitations of the parameterisations relied upon are clearly stated. The results and conclusions are discussed in context of existing literature throughout.

**Significance**

Whilst the motivation for the study set out to disentangle the drivers of Sermeq Kujalleq's observed retreat between 1985 and 2018, and the results indicate a better understanding of these drivers, the study also illustrates the significant role that melange may play in buttressing glaciers and inhibiting calving. This is currently an aspect of ice sheet modelling with limited understanding and despite limitations in the approach taken in this study, the authors highlight how the community may benefit from future research in this area.

However, an aspect that is currently not discussed in the manuscript is the suitability of employing the tensile stress based calving law (or von Mises calving law (Morlighem et al 2016). Although this calving law has been used relatively frequently in the literature over recent years, it is not so clear whether this calving law best parameterises the physical processes that drive calving. That is due to the dependence firstly on the tuning parameter (here the threshold stress parameter), and secondly the dependence of the derived calving rate on the ice flow velocity when the two are not necessarily related.

The reason for mentioning the specific calving law here is that it is unclear whether or not the conclusions of the study are robust. For example, if a different calving law was employed, would the melt and calving fluxes be reproduced in the same way discussed here and would such a dependence on the melange backstress be required to reproduce the observed calving rates. The purpose of the study was not to compare calving laws however, and considering ways to include parameterisations of melange backstress is an important feature of the study which may encourage further research in this area.

**Presentation Quality**

The methods, results and conclusions of the study are presented clearly and the manuscript flows well. The figures contribute well to the understanding of the study and on the whole can be interpreted easily. Some results and key findings of the study rely on results which are only presented in the supplementary data – for example, it is important to understand the near-terminus stresses in the glacier in the context of the varying stress threshold which controls the calving rate. The comparison of the change in flux due to melt and calving is nice, but has also been kept in the supplementary data. These figures being in the main text would add to the readers' interpretation of results.

**Reviewer comments and manuscript updates**

After consideration of reviewer comments, the authors have made significant modifications to the manuscript which have added greater clarity to the methodologies undertaken and interpretation of results within the context of current literature. The authors are open about the limitations of the study and highlight where simplified parameterisations have been employed to represent physical processes which are currently not well understood.

---

## Referee Report (RR2)

**Review ´Disentangling the drivers behind the post-2000 retreat of Sermeq Kujalleq, Greenland (Jakobshavn Isbrae)´**

This study examines the retreat of the Sermeq Kujalleq glacier from 1985 to 2018, utilizing ensemble modeling to assess the role of frontal and submarine melt, as well as calving. Unlike previous studies, this research adapts the calving law to incorporate the backstress effect of ice mélange. To predict the glacier's future evolution, understanding the relative importance of these drivers is crucial. While the study cannot fully answer all open questions, it offers a thoughtful discussion of potential scenarios. Some assumptions and simplifications in the model could be better justified to enhance the interpretation of the results. With a few minor revisions to the text and figures, the manuscript would provide a valuable contribution to understanding the dynamics of Sermeq Kujalleq.

Comments

As you are looking only at oceanic drivers, I would clearly state this in the title.

Abstract

Highlighting more what your study is adding to the model compared to former studies and which results are new would help to show the importance of your study. For example L10-13 is already known and could be shortened but then an explanation why you conclude that ´more sophisticated models of iceberg mélange and calving evolution´ are needed would be useful.

Introduction

L18: Sermeq Kujalleq is not the only Sermeq Kujalleq in Greenland. For clarification add Sermeq Kujalleq in Kangia here.

L30-31: I can understand your reasoning of comparability, but it seems to rather belong to the method section. Here you can just say that the collapse of the ice tongue is widely believed to have initiated the retreat although the study of Gladish et al. suggested that the warmer water did not reach SK.

L32: In the sentence before you write that the warmer water did not reach SK and now it is generally agreed that warm subsurface water triggered the retreat. Rephrase to make these two sentences not contradicting each other.

L34-38: Formulated like this, it doesn't fit into this section of the introduction. You can move this part to the last part of the introduction, where you explain what you are doing in your study. Here I would just explain what the different physical processes are and potentially link to recent studies.

L39-40: A delay or another mechanism responsible for the increase in flow speed?

L63: For SK not just in winter, also in summer it is not possible to collect ocean measurements in proximity of the glacier terminus. But other close-range observations are possible and available (seismic, terrestrial radar, drones; e.g. Xie et al. 2016). However, they are often restricted to shorter time periods. Additionally, satellite observations can help to understand for example ice mélange density (e.g. Wehrlé et al., 2023). Observations are still rare but increasingly available (e.g. Kim et al., 2024, Wehrlé et al. 2023; Xie et al. 2016, 2018).

Methods:

L156: Do you have any evidence (e.g. observations) supporting this linear sensitivity?

L185-187: What if subglacial meltwater discharge would increase a lot? Several recent studies show increased calving activity in the proximity of convective plumes. I understand that you are not considering convective plumes due to complexity, but I am wondering if this assumption holds true with increased melt.

L191: I do not understand your justification here. Any evidence?

Figure 1: It is hard to see which dashed line corresponds to which solid line. Please change the colors.

L212: I miss the source of your calving front location catalogue.

Discussion:

L395-397: This sentence is a bit confusing ('insufficiently strong ocean melt is unable..'), please rephrase.

L396: stong -> strong

L397: Within these simulations -> Simulations with insufficient ocean melt?

L405: Could there be other reasons than the simplified parametrization of ice mélange?

L408-410: Could you shortly mention which simplifications limit the applicability? Or is this related to the next sentences?

L431: However -> However

Conclusion:

How robust does the ice mélange has to be to suppress calving activity and is this realistic?

Supplemental material:

There are grounding line positions for summer 2016 available (Kim et al., 2024). Do they agree with your modelled grounding line positions?

---

## Author Response (AR2)

**Review ´Disentangling the drivers behind the post-2000 retreat of Sermeq Kujalleq, Greenland (Jakobshavn Isbrae)´**

This study examines the retreat of the Sermeq Kujalleq glacier from 1985 to 2018, utilizing ensemble modeling to assess the role of frontal and submarine melt, as well as calving. Unlike previous studies, this research adapts the calving law to incorporate the backstress effect of ice mélange. To predict the glacier's future evolution, understanding the relative importance of these drivers is crucial. While the study cannot fully answer all open questions, it offers a thoughtful discussion of potential scenarios. Some assumptions and simplifications in the model could be better justified to enhance the interpretation of the results. With a few minor revisions to the text and figures, the manuscript would provide a valuable contribution to understanding the dynamics of Sermeq Kujalleq.

**Thank you for your thoughtful comments. We have addressed the issues raised below.**

Comments

As you are looking only at oceanic drivers, I would clearly state this in the title.

**Changed**

Abstract

Highlighting more what your study is adding to the model compared to former studies and which results are new would help to show the importance of your study. For example L10-13 is already known and could be shortened but then an explanation why you conclude that ´more sophisticated models of iceberg mélange and calving evolution´ are needed would be useful.

**This would be a lot of explanation for the abstract. We have opted to keep it as is to ensure readability.**

Introduction

L18: Sermeq Kujalleq is not the only Sermeq Kujalleq in Greenland. For clarification add Sermeq Kujalleq in Kangia here.

**Corrected to Sermeq Kujalleq in Kangia**

L30-31: I can understand your reasoning of comparability, but it seems to rather belong to the method section. Here you can just say that the collapse of the ice tongue is widely believed to have initiated the retreat although the study of Gladish et al. suggested that the warmer water did not reach SK.

**The late 1990s saw warmer subsurface waters arrive in Disko Bay and the Illulisat Icefjord, leading to the collapse of the floating ice tongue, which is widely believed to have initiated SK's retreat and acceleration over the next 20 years, although \citet{Gladish2015} suggest that warmer waters may not have reached SK's front due to the downstream sill.**

L32: In the sentence before you write that the warmer water did not reach SK and now it is generally agreed that warm subsurface water triggered the retreat. Rephrase to make these two sentences not contradicting each other.

**Corrected the contradictory statement, by including that Gladish (2015) provides a reason for intrusion into fjord by overcoming the sill. This occurred due to the rising Irminger**

**Water layer as a byproduct of cyclonic circulation in Disko Bay.**

L34-38: Formulated like this, it doesn't fit into this section of the introduction. You can move this part to the last part of the introduction, where you explain what you are doing in your study. Here I would just explain what the different physical processes are and potentially link to recent studies.

**Reformatted to address**

L39-40: A delay or another mechanism responsible for the increase in flow speed?

**Corrected in text**

L63: For SK not just in winter, also in summer it is not possible to collect ocean measurements in proximity of the glacier terminus. But other close-range observations are possible and available (seismic, terrestrial radar, drones; e.g. Xie et al. 2016). However, they are often restricted to shorter time periods. Additionally, satellite observations can help to understand for example ice mélange density (e.g. Wehrlé et al., 2023). Observations are still rare but increasingly available (e.g. Kim et al., 2024, Wehrlé et al. 2023; Xie et al. 2016, 2018).

**Added additional references**

Methods:

L156: Do you have any evidence (e.g. observations) supporting this linear sensitivity?

**Robel, 2017 argues that buttressing strength is linearly dependent on mélange thickness. We do not have observations relating mélange strength to temperature, however we use temperature as a proxy for mélange robustness to assume a linear stress-temperature calving threshold relation.**

L185-187: What if subglacial meltwater discharge would increase a lot? Several recent studies show increased calving activity in the proximity of convective plumes. I understand that you are not considering convective plumes due to complexity, but I am wondering if this assumption holds true with increased melt.

**It could, but this consideration is beyond the scope of this study.**

L191: I do not understand your justification here. Any evidence?

**We match the nearest point in time modeled calving front to our observed calving front**

Figure 1: It is hard to see which dashed line corresponds to which solid line. Please change the colors.

**Corrected in text**

L212: I miss the source of your calving front location catalogue.

**Corrected in text**

Discussion:

L395-397: This sentence is a bit confusing ('insufficiently strong ocean melt is unable..'), please rephrase.

**Corrected in text**

L396: stong -> strong

**Corrected in text**

L397: Within these simulations -> Simulations with insufficient ocean melt?

**Corrected in text**

L405: Could there be other reasons than the simplified parametrization of ice mélange?

**Yes there could be, but we highlight that mélange presence is the dominant control on calving activity at SK.**

L408-410: Could you shortly mention which simplifications limit the applicability? Or is this related to the next sentences?

**It is related to the next sentences**

L431: However -> However

**Fixed in text**

Conclusion:

How robust does the ice mélange has to be to suppress calving activity and is this realistic?

**Since we do not physically model a buttressing stress we cannot sufficiently answer this question. However, SK's mélange has been shown to have a direct control on the stability of SK's calving front. Amundson, 2010 argues that a mélange buttressing stress on the order of 100-200kPA can prevent or decelerate an overturning iceberg during calving.**

Supplemental material:

There are grounding line positions for summer 2016 available (Kim et al., 2024). Do they agree with your modelled grounding line positions?

**Yes, similarly to (Kim et al., 2024), our simulated SK's grounding line arrests at a pinning point the southern trunk. We note in text that grounding lines which recede past this point of arrest continue to experience unabated retreat.**